# Novel Approaches in Chronic Renal Failure without Renal Replacement Therapy: A Review

**DOI:** 10.3390/biomedicines11102828

**Published:** 2023-10-18

**Authors:** Sandra Luz Martínez-Hernández, Martín Humberto Muñoz-Ortega, Manuel Enrique Ávila-Blanco, Mariana Yazmin Medina-Pizaño, Javier Ventura-Juárez

**Affiliations:** 1Departamento de Microbiología, Centro de Ciencias Básicas, Universidad Autónoma de Aguascalientes, Aguascalientes 20100, Ags, Mexico; 2Departamento de Química, Centro de Ciencias Básicas, Universidad Autónoma de Aguascalientes, Aguascalientes 20100, Ags, Mexico; 3Departamento de Morfología, Centro de Ciencias Básicas, Universidad Autónoma de Aguascalientes, Aguascalientes 20100, Ags, Mexico

**Keywords:** chronic kidney disease, anti-fibrotic drugs, new diuretics drugs, novel anti-hypertensives drugs, metabolism-regulating drugs

## Abstract

Chronic kidney disease (CKD) is characterized by renal parenchymal damage leading to a reduction in the glomerular filtration rate. The inflammatory response plays a pivotal role in the tissue damage contributing to renal failure. Current therapeutic options encompass dietary control, mineral salt regulation, and management of blood pressure, blood glucose, and fatty acid levels. However, they do not effectively halt the progression of renal damage. This review critically examines novel therapeutic avenues aimed at ameliorating inflammation, mitigating extracellular matrix accumulation, and fostering renal tissue regeneration in the context of CKD. Understanding the mechanisms sustaining a proinflammatory and profibrotic state may offer the potential for targeted pharmacological interventions. This, in turn, could pave the way for combination therapies capable of reversing renal damage in CKD. The non-replacement phase of CKD currently faces a dearth of efficacious therapeutic options. Future directions encompass exploring vaptans as diuretics to inhibit water absorption, investigating antifibrotic agents, antioxidants, and exploring regenerative treatment modalities, such as stem cell therapy and novel probiotics. Moreover, this review identifies pharmaceutical agents capable of mitigating renal parenchymal damage attributed to CKD, targeting molecular-level signaling pathways (TGF-β, Smad, and Nrf2) that predominate in the inflammatory processes of renal fibrogenic cells.

## 1. Introduction

Chronic kidney disease (CKD) encompasses a complex array of etiological factors that can culminate in severe complications, notably renal failure [1]. It is imperative to underscore the intimate connection between acute kidney disease (ACKD) and CKD, with the former often acting as a precipitating factor for the latter, especially among elderly patients. Therefore, a meticulous assessment of renal function in patients presenting with ACKD is paramount, along with the implementation of judicious therapeutic strategies to avert the progression to CKD. An investigation conducted in 2017 [2] emphasizes that the prevalence of ACKD escalates with advancing age, placing elderly individuals at heightened susceptibility because of age-related physiological alterations, including reduced renal blood flow and declining renal function [3]. CKD is characterized by structural or functional abnormalities within the kidney, or a persistent decline in the glomerular filtration rate (GFR) < 60 mL/min/1.73 m^2^ [4,5], extending for a minimum duration of ≥3 months. It is categorized into five gradations: (1) Grade 1, denoting normal renal function with a GFR > 90 mL/min/1.73 m^2^; (2) Grade 2, signifying renal impairment with a mild reduction in GFR to 60–89 mL/min/1.73 m^2^; (3) Grade 3, representing a moderate reduction in GFR, further subcategorized into 3a (mildly reduced) with a GFR of 45–59 mL/min/1.73 m^2^ and 3b (moderately to severely reduced) with a GFR of 30–44 mL/min/1.73 m^2^; (4) Grade 4, indicating a severely reduced GFR at 15–29 mL/min/1.73 m^2^; and (5) Grade 5, signifying renal insufficiency (renal failure) with a GFR < 15 mL/min/1.73 m^2^ [6]. The most prevalent etiologies encompass diabetes mellitus (DM) and arterial hypertension (AH) [7]. Furthermore, factors such as infectious glomerulonephritis, renal vasculitis, ureteral obstruction, genetic mutations, autoimmune diseases (immunoglobulin A nephropathy, lupus nephritis, and Goodpasture’s disease) [8,9], and pharmaceutical agents [10,11] can instigate CKD. Renal damage may be exacerbated by predisposing factors or other elements that expedite the deterioration of renal function, consequently elevating morbidity and mortality rates in CKD patients [12]. In 2015, CKD ranked as the 17th leading cause of global disability and the 12th leading cause of mortality worldwide, accounting for 1.1 million fatalities. Over the past decade, mortality rates have witnessed a noteworthy 31.7% escalation worldwide. Consequently, CKD has evolved into an exigent public health concern, primarily attributed to its pervasive prevalence and the formidable global expenditures associated with its treatment and ensuing complications. In 2020, the University of Washington, within its Global Health Metrics report [13], disseminated data regarding the incidence, prevalence, and mortality of diseases and injuries impacting the global populace. Notably, CKD emerged as the sixth leading cause of mortality, encompassing both men and women aged 19–64, within a global context. Simultaneously, primary contributors to CKD, such as AH and DM, were documented as the first and third leading causes of mortality within the adult population, respectively [13,14]. These findings underscore the pivotal role CKD plays as a preeminent global cause of mortality. The overarching objectives of CKD management, particularly within its non-renal function replacement grades (G2–G4), entail stringent dietary control, precise regulation of mineral salt levels, and meticulous management of blood pressure, blood glucose, and fatty acid levels. Nonetheless, the currently available therapeutic modalities for CKD have exhibited limited efficacy in halting the relentless progression of renal damage. Hence, this comprehensive investigation embarks on a discerning exploration of innovative therapeutic paradigms, aimed at attenuating the pervasive inflammation, mitigating extracellular matrix accrual, and fostering the regeneration of compromised renal tissue in the complex landscape of CKD. To facilitate this extensive inquiry, we executed a nonsystematic review utilizing the PubMed Central database. Employing the search parameters “Chronic, And Kidney, And Disease, And Treatment, And Adult”, our search spanned the period from 2010 to 2023, yielding an initial corpus of approximately 32,000 pertinent papers. Exclusion criteria were meticulously applied, excising articles centered on renal function replacement therapy, ultimately culminating in the inclusion of a total of 240 papers that form the foundation of this comprehensive review.

## 2. Main Etiologies of CKD

### 2.1. Arterial Hypertension

In physiological conditions, the kidneys maintain intraglomerular pressure through the regulation of afferent and efferent arteriole tone to preserve glomerular filtration rate (GFR) and urine flow. However, in cases of decreased renal perfusion pressure or volume depletion, the synthesis and release of various mediators, notably prostaglandins E2 and I2, kinins, nitric oxide, and atrial natriuretic peptides, increase. Prostaglandins E2 (PGE2) and I2 (PGI2) in the kidneys exert an antihypertensive effect through vasodilation, which is counteracted by nonsteroidal anti-inflammatory drugs (NSAIDs). The exact role of kinins in CKD remains incompletely understood; nevertheless, their stimulation of pain may be linked to prostaglandins, and vasodilation may result from nitric oxide release upon binding to the B2 receptor [15].

Intraglomerular pressure is primarily regulated through the classical angiotensin II pathway, where the renin–angiotensin system (RAS) activation leads to preferential vasoconstriction of the efferent arteriole. However, mounting evidence suggests the existence of numerous other angiotensin peptide products generated by endo- and carboxypeptidases, including angiotensin III (Ang 2–8), Ang 1–7, Ang 1–9, Ang 3–7, and Ang 3–8, each exerting distinct physiological effects [16,17]. Angiotensin II, a key player, increases intraglomerular pressure, a pivotal factor in CKD progression. Renal effects of angiotensin II and aldosterone encompass the stimulation of mesangial cell mitogenesis, local inflammation, and fibrosis [18]. Consequently, correcting aberrations in the renin–angiotensin–aldosterone system (RAAS) has emerged as a foremost therapeutic approach to decelerate CKD progression, often achieved by angiotensin-converting enzyme (ACE) inhibitors and angiotensin II receptor blockers, either individually or in combination [18,19]. Angiotensin II has been identified to directly induce apoptosis in podocytes, the third layer of cells forming the glomerular filtration barrier, by upregulating the Na+/H+ exchanger type 1 (NHE-1) protein—a critical intracellular pH regulator correlating with the production of reactive oxygen species (ROS). This process can be prevented by the administration of olmesartan and amiloride, the latter being a potassium-sparing diuretic and NHE-1 inhibitor. In the alternative RAS pathway, angiotensin (1–7) assumes a central role, generating nitric oxide by binding to the G-protein-coupled receptor related to Mas type D (MrgD), resulting in vasodilation and conferring a reno-protective effect [20]. Furthermore, Yu L. et al. (2016) [21] have demonstrated that the Ang 1–5 metabolite exhibits cardioprotective properties, stimulating the release of cardioprotective atrial natriuretic peptide (ANP) through the Mas axis, like its parent peptide Ang 1–7. Several other mediators, including adenosine, endothelin 1, vasopressin (antidiuretic hormone), noradrenaline, thromboxane A2, and leukotrienes D4 and C4, also act as renal vasoconstrictors, often working in concert with angiotensin II [15].

### 2.2. Diabetes Mellitus

Diabetes mellitus (DM) constitutes a cluster of metabolic disorders characterized by hyperglycemia resultant from anomalies in insulin secretion, insulin responsiveness, or a combination thereof. Prolonged hyperglycemia is intrinsically linked to enduring organ damage and dysfunction, with a pronounced predilection for the ocular, renal, neural, cardiovascular, and vascular systems. Among the microvascular complications prevalent in both type 1 (T1DM) and type 2 (T2DM), diabetic nephropathy emerges as the most frequent. This condition is typified by persistent proteinuria, progressive renal function deterioration, glomerular hypertrophy, mesangial expansion, thickening of the glomerular basement membrane, and interstitial fibrosis. Globally, it stands as a foremost contributor to end-stage renal disease, affecting over 20% of individuals afflicted with DM [22].

These vascular complications exhibit a strong correlation with endothelial dysfunction brought about by oxidative stress resulting from hyperglycemia [23]. Consequently, mesangial expansion, a pivotal element in the genesis of diabetic nephropathy, ensues, accompanied by the abnormal proliferation of mesangial cells and the accumulation of matrix proteins within the glomerular core. Hence, mesangial expansion serves as a prominent diagnostic criterion for clinical evaluation of diabetic nephropathy [24]. On a parallel note, hyperglycemia typically escalates the concentration of advanced glycation end products (AGEs) and reactive oxygen species (ROS), formed through nonenzymatic glycation reactions involving reducing sugars, amino acids, lipids, or DNA [25].

#### 2.2.1. Microangiopathic Complications

Elevated levels of circulating glucose in the bloodstream can redirect this glucose into alternative biochemical pathways, resulting in an increased production of advanced glycation end products (AGEs), autooxidation of glucose, flux of hexosamines and polyols, and activation of classical isoforms of protein kinase C. These molecular events are considered pivotal mediators of cellular injury induced by hyperglycemia. Multiple pathways implicated in hyperglycemia-induced endothelial dysfunction converge to yield a substantial generation of reactive oxygen species (ROS), thereby precipitating oxidative stress. Excess ROS can intensify cellular damage by triggering a cascade of biochemical reactions that continue ROS generation in response to hyperglycemic conditions, thus establishing a self-perpetuating cycle. The superoxide anion (O_2_^−^) can react with nitric oxide (NO) to yield peroxynitrite, which is recognized as a contributor to endothelial dysfunction. The origins of the endoplasmic reticulum (ER) stress response remain enigmatic, whether it arises from a direct response to heightened protein synthesis and maturation because of hyperglycemia or is induced by oxidative stress resulting from hyperglycemic conditions. It is plausible that the ER stress response might also contribute to the excessive production of ROS, thereby introducing a potentially intricate interplay between cellular stress responses [23].

#### 2.2.2. Formation of Late Glycation End Products

Advanced glycation end products (AGEs) constitute a chemically diverse and heterogeneous group of compounds that form within the human body through both endogenous and exogenous pathways. These AGEs originate from nonenzymatic processes involving the condensation of carbonyl groups found in reducing sugars with free amino groups present in nucleic acids, proteins, or lipids. This chemical interaction leads to the creation of stable and irreversible end products. Over recent decades, AGEs have gained significant prominence in the scientific community because of mounting evidence implicating their involvement in a wide range of physiopathological processes and diseases. These encompass conditions spanning from diabetes mellitus (DM), cancer, and cardiovascular disorders to neurodegenerative diseases. Notably, their relevance has even extended to diseases such as SARS-CoV-2 viral infection. AGEs interact with a variety of cellular receptors, initiating intricate signaling cascades closely associated with inflammatory responses and oxidative stress [26].

#### 2.2.3. Activation of the Hexosamine Pathway

Glycosylation entails the utilization of hexosamines derived from nutrients such as glucose, glutamine, and pyrimidines. This process plays a pivotal role in remodeling N-glycans within maturing glycoproteins within the Golgi apparatus, as well as in O-linked glycosylation (Ser/Thr) of various signaling proteins. Dysfunctional N-glycosylation of proteins can lead to misfolding, subsequently triggering endoplasmic reticulum (ER) stress. Impaired expression and/or glycosylation of cell surface proteins disrupt cellular signaling pathways, thereby exerting a significant influence on cellular fate. Anomalous lipid glycosylation intricately impacts organelle biogenesis, trafficking, and signaling processes [27]. The hexosamine biosynthesis pathway (HBP) represents a relatively minor aspect of glycolysis, accounting for approximately 2–5% of total glucose metabolism. However, under conditions of hyperglycemia, this pathway can induce post-translational protein modifications through glycosylation and the synthesis of glycolipids, proteoglycans, and glycosylphosphatidylinositol anchors. In this pathway, fructose-6-phosphate is converted to glucosamine-6-phosphate, catalyzed by the initial enzyme glutamine: fructose-6-phosphate amido transferase (GFAT). This process primarily yields UDP-N-acetylglucosamine (UDP-GlcNAc), a crucial substrate for various reactions, including proteoglycan synthesis and the formation of O-linked glycoproteins. Excessive modification by glucosamine often results in pathological shifts in gene expression closely linked to specific metabolic consequences of prolonged hyperglycemia, thereby promoting diabetes-related complications. The reversible modification of protein O-GlcNAcylation, triggered by heightened HBP activity, may potentially lead to insulin resistance and diabetic complications. Of particular relevance to diabetic vascular complications is the inhibition of endothelial nitric oxide synthase (e-NOS) activity in arterial endothelial cells because of O-GlcNAcylation [28]. We have elucidated the principal etiologies of CKD. Nevertheless, this condition typically initiates a chronic inflammatory cascade characterized by fibrotic changes in renal tissue, culminating in the physiological manifestation of chronic renal insufficiency [29].

### 2.3. Pathophysiology of Fibrosis

Renal fibrosis arises because of an unsuccessful attempt at tissue repair following initial insults. Prolonged or chronic renal injury stimulates the production of various cell types, including fibroblasts, tubular epithelial cells, pericytes, endothelial cells, vascular smooth muscle cells, mesangial cells, and podocytes, in addition to circulating cells such as lymphocytes, macrophages, and fibrocytes [29,30,31]. There is also evidence suggesting the involvement of renal stellate cells in renal fibrosis [32]. Clinically, renal fibrosis presents as glomerulosclerosis, tubulointerstitial fibrosis, inflammatory infiltration, and the loss of renal cells. The damaged renal cells are gradually replaced by the extracellular matrix (ECM) [33,34,35,36,37,38]. This process leads to tubular atrophy, capillary leakage, depletion of podocytes through reactive oxygen species (ROS)-mediated mechanisms, and the release of inflammatory and fibrogenic cytokines, ultimately resulting in the disruption of the normal renal architecture [39,40]. This disruption includes the aggregation of collagen fibers, functional deterioration, and the activation of myofibroblasts via transforming growth factor beta (TGF-β), involving mesangial cells, fibroblasts, and epithelial-mesenchymal transition [34,41,42,43,44,45,46]. The TGF-β signal is transduced through serine/threonine kinase type I and II receptors [47], leading to the production and release of fibrogenic factors such as TGF-β1, platelet-derived growth factor, fibroblast growth factor 2 (FGF2), connective tissue growth factor, angiotensin II [48], Notch1 [49,50], and hypoxia-inducible factor 1 [51,52]. These factors play critical roles as intracellular signaling mediators [35,36,37,38].

## 3. CKD Treatment

### 3.1. General Considerations

Medical treatment for CKD is tailored to the clinical indicators of the primary etiology. Commencing with outpatient care, the diagnostic process relies on renal function tests, encompassing parameters such as urea, creatinine, and estimated glomerular filtration rate (eGFR). CKD typically presents as asymptomatic, and when symptoms do manifest, they commonly encompass general discomfort, loss of appetite, and disturbances in sleep patterns. These manifestations are particularly prevalent among individuals with AH or T2DM. It is noteworthy that concurrent conditions linked to AH or T2DM, such as atherosclerosis, warrant careful consideration because of their significant roles as predisposing factors in the development of CKD [53]. Findings from the National Observatory Study of Atherosclerosis in Nephrology (NEFRONA) highlight the heightened risk of atheromatous plaque formation in CKD patients, as evidenced by a 68.7% prevalence based on carotid and femoral ultrasound scans. Moreover, a significant correlation has been established between CKD stages 3–5 and reduced levels of 25-OH Vitamin D [54]. CKD is further linked to obesity and is characterized by glomerulomegaly, frequently accompanied by localized and segmental glomerulosclerosis lesions. In this patient cohort, initial symptoms tend to deviate from the typical, with microproteinuria emerging as the principal clinical presentation [55]. Furthermore, emphasizing the role of nicotine in exacerbating the progression of CKD, both clinical and experimental evidence strongly suggest that nicotine, to a considerable extent, contributes to the detrimental impact of cigarette smoking on CKD progression [56]. Nicotine promotes renal interstitial fibrosis via the up-regulation of XIAP in an alpha7-nAChR-dependent manner [57].

The kidney, as the principal organ responsible for xenobiotic elimination, is particularly susceptible to the toxic effects of pharmaceuticals and their metabolites during therapeutic interventions. The excretion of drugs and their metabolites also occurs via urine, predominantly through processes of glomerular filtration and tubular secretion. A variety of nephrotoxic drugs, including ACE inhibitors, ARBs, NSAIDs, mitomycin-C, antiplatelet agents, cyclosporin, quinone compounds, aminoglycosides, amphotericin B, adefovir, cisplatin, foscarnet, contrast media, cocaine, heroin, methadone, methamphetamines, lithium, acyclovir, indinavir, and sulfonamides, have been implicated in drug-induced kidney disease (DIKD), affecting a substantial proportion of adult (14–26%) and pediatric (16%) patients. However, the conventional measurement of nephrotoxicity using serum creatinine or BUN lacks the requisite sensitivity and specificity to detect renal injury in its early stages before progressing to severe damage. Biomarkers play a pivotal role in advancing drug development by facilitating the assessment of drug toxicity. A diverse array of biomarkers, including albumin, transferrin, immunoglobulin G, β2-microglobulin, α1-microglobulin, cystatin C, retinol-binding protein, interferons, interleukins, TNF, CSFs, type IV collagen, α-GST, N-Acetyl-D-Glucosaminidase, KIM-1, NGAL, clusterin, and osteopontin, contribute significantly to this evaluation process [58,59]. Diagnosis may also be established by identifying direct manifestations of kidney damage, such as oedema, nausea, vomiting, muscle discomfort, and anemia. Upon diagnosis, a comprehensive medical management strategy is initiated, necessitating a multidisciplinary approach involving nephrologists, nutritionists, renal nurses, and social workers. Central to this approach is the establishment of effective communication channels among the medical team, the patient, and the patient’s family. These communication channels serve as a critical means to facilitate conservative outpatient treatment, especially when patients fall within CKD stages 2–4, with the goal of averting the need for renal replacement therapy [6,60].

### 3.2. Antihypertensives

A wealth of scientific investigations has substantiated the dual role of AH as both a causative factor and a complicating condition in CKD. Hypertension causes functional and structural alterations within the kidneys and stands as a significant risk factor for cardiovascular complications (A Chinese Nationwide Cohort Study in Taiwan, 2001–2008, Clinical Trials Identifier NCT01593787) [61,62]. Among the therapeutic arsenal for CKD, ACE inhibitors and angiotensin II receptor antagonists stand as fundamental pillars. The efficacy of this therapeutic approach in retarding CKD progression has been validated by multiple clinical trials. Remarkably, within the United States, approximately 72% of individuals afflicted with CKD and type 2 diabetes mellitus (T2DM) receive treatment with ACE inhibitors, cementing their status as the second most widely prescribed class of antihypertensive medications. These pharmaceutical agents find utility not only in blood pressure management but also in the prevention of cardiovascular disease and CKD (Brigham and Women’s Hospital and Massachusetts General Hospital, 2009–2011, CRIC Study 2003–2008) [63,64,65]. Furthermore, the exploration of the renin–angiotensin–aldosterone system (RAAS) has led to the identification of several drugs designed to selectively inhibit pathways that contribute to the hypertensive cascade, mirroring the mechanism of action of angiotensin II type 1a receptor blockers (as delineated in Table 1) [64,66].

#### 3.2.1. ACE Inhibitors

The mechanism of action of ACE inhibitors centers on the inhibition of ACE, an enzyme responsible for catalyzing the conversion of angiotensin I to angiotensin II. Angiotensin-producing sites are distributed within the liver, spleen, and bronchial tissues [66]. ACE inhibitors serve as nephroprotective agents that mitigate cardiovascular morbidity and mortality. Notable ACE inhibitors employed in contemporary clinical practice encompass benazepril [67], captopril [68], enalapril [69], fosinopril [70], lisinopril [71], perindopril [72], ramipril [73], and others [67,74]. Nonetheless, there exists a dearth of published clinical trials assessing the efficacy and safety profiles of ACE inhibitors in individuals with advanced CKD (CRIC, Chronic Renal Insufficiency Cohort study, June 2003-September 2008) [64,66]. In a nationwide observational study involving individuals with advanced CKD, discontinuing RAS inhibition was associated with heightened absolute mortality risks, increased incidence of major adverse cardiovascular events, and a decreased absolute risk of initiating kidney replacement therapy (Swedish Renal Registry, 2007–2017) (see Table 1) [61,64,118].

#### 3.2.2. Angiotensin Receptor Blockers (ARBs)

Unlike ACE inhibitors, angiotensin receptor blockers (ARBs) do not inhibit the conversion of angiotensin I to angiotensin II. Instead, they function by obstructing the binding of angiotensin type II receptors, which are prevalent in smooth muscle tissue, adrenal glands, myocardium, and notably, within the renal glomeruli [77]. An additional mechanism entails the blockade of aldosterone-mediated release, making aldosterone-blocking drugs a valuable asset in clinical practice. ARBs assume a pivotal role in the management of various conditions, including arterial hypertension (AH), heart failure, ACE inhibitor intolerance, proteinuria in CKD, diabetic nephropathy, post-myocardial infarction, and proteinuria in post-transplant kidney recipients [77,81]. Numerous meta-analyses have underscored their antihypertensive and antiproteinuric properties [77,119,120,121,122]. Losartan, as the first ARB extensively studied, has been instrumental in elucidating their efficacy in mitigating renal damage in individuals with T2DM and HA. A comparative analysis of losartan and amlodipine (a calcium channel blocker) has demonstrated the efficacy of both agents in achieving sustained blood pressure reduction over a 24 h period [61,120]. Moreover, ARBs exhibit additional benefits, such as the prevention of ventricular remodeling, inhibition of proinflammatory cytokine production (including TNF-α, IL-6, chemotactic protein, resistin, and tissue plasminogen activator), and pleiotropic effects not demonstrated by other drug classes such as calcium antagonists, diuretics, and beta-blockers [61,77,119,121,123]. An analysis encompassing 411 patients with advanced CKD, randomly assigned to discontinue or continue renin–angiotensin system (RAS) inhibitors or ARBs, advanced or delayed renal replacement therapy, respectively. A group of patients with a mean eGFR of 12.6 ± 0.7 mL/min/1.73 m^2^ had their treatment interrupted, while another group with a mean eGFR of 13.3 ± 0.6 mL/min/1.73 m^2^ had their treatment continued. Heterogeneity was not observed in the prespecified subgroups. The initiation of renal replacement therapy occurred in 128 patients (62%) in the discontinuation group and 115 patients (56%) in the continuation group. Adverse events were similar in both groups with respect to cardiovascular events (108 vs. 88) and deaths (20 vs. 22). Among patients with advanced and progressive CKD, discontinuing RAS inhibitors did not significantly correlate with the rate of long-term decline in eGFR (Table 1), demonstrating that discontinuation advanced the initiation of renal replacement therapy, while continuation delayed it. Notably, the two groups showed comparable adverse events, including cardiovascular events and mortality [124] (see Table 1).

#### 3.2.3. Renin Inhibitors

Given that renin is a significant vasoactive agent with vasoconstrictive properties and multifaceted effects at various levels, the hypothesis has been raised that these inhibitors may exhibit greater efficacy than ACE inhibitors and ARBs. However, it is important to note that there is a lack of evidence supporting this hypothesis [19]. Renin inhibitors show promise as agents for the treatment of hypertension, and their characteristics differ from those of ARBs. For instance, experimental studies have demonstrated a reduction in the size of fibrotic renal lesions in animal models [19,77,80]. Furthermore, the combination of aliskiren (a renin inhibitor) with losartan has shown observable nephroprotective effects in patients with type 2 diabetes mellitus and nephropathy, regardless of the hypotensive effect generated (Table 1) [19,77,80,121].

#### 3.2.4. Calcium Channel Blockers

Compelling evidence suggests that non-dihydropyridine calcium channel blockers (e.g., verapamil and diltiazem) can reduce proteinuria in individuals with renal impairment. A multicenter trial involving patients with T2DM and nephropathy suggested that the combination of a non-dihydropyridine calcium channel blocker and an ACE inhibitor might lead to a reduction in albuminuria, potentially accompanied by a decrease in systolic blood pressure. This trial adhered to the principles of a prospective, randomized, open-label, blinded end-point (PROBE) design [77,78,79]. Among the long-acting calcium channel blockers, such as azelnidipine and lercanidipine (but not amlodipine), there is evidence of improved renal outcomes in patients with chronic renal damage [78,125]. In contrast, cilnidipine, a calcium channel blocker, modulates L-type calcium channels in vascular smooth muscle cells and exerts antihypertensive effects akin to amlodipine on N-type calcium channels within sympathetic nerve endings. This unique dual activity suggests that cilnidipine, when combined with RAS inhibitors, could serve as an effective antihypertensive medication that not only inhibits the progression of CKD but also mitigates associated cardiovascular complications (see Table 1) [126].

#### 3.2.5. Adrenoblockers

Beta blockers such as celiprolol, carvedilol, and nebivolol offer advantages to patients with CKD by providing vasodilatory properties without withdrawal side effects. Notably, bisoprolol has been observed to accumulate in CKD patients, leading to adverse outcomes, and as a result, its use is not recommended [70]. However, it is important to note that in countries like the United Kingdom, beta blockers are not typically considered as first-line antihypertensive agents for the general population. Nevertheless, evidence underscores that patients with type 2 diabetes mellitus (T2DM) and advanced CKD face an elevated risk of sudden death. In light of this, the capacity of beta blockers to reduce heart rate, mitigate sympathetic hyperactivity, and prevent ventricular arrhythmias positions them as effective therapeutic options. A meta-analysis focusing on beta blockers for CKD lends support to their utility, particularly in patients concurrently experiencing heart failure (see Table 1) [77,81].

### 3.3. Diuretics

Diuretics are widely used pharmacological agents in routine clinical practice. All diuretics, except spironolactone, eplerenone, and a new nonsteroidal mineralocorticoid blocker, finerenone, must reach the luminal space to exert their effects (STOP-CKD, ISRCTN80658312, 4 April 2013–29 February 2016) [83]. The entry of diuretics into the urinary compartment is influenced by the glomerular filtrate because of the extensive binding of diuretics to proteins [76,83,84,127]. The mineralocorticoid receptor antagonist spironolactone has been found to significantly reduce arterial stiffness, left ventricular mass, and proteinuria and to delay CKD progression (BARACK D, EudraCT: 2012-002672-13, 28 June 2013–17 October 2022) [83,84]. A meta-analysis conducted in 2009 and updated in 2014 demonstrated that the addition of spironolactone to ARB therapy reduces blood pressure and proteinuria in CKD patients ((1966–2014), EMBASE (1947–2014), and the Cochrane Clinical Trials Database) [127]. Furosemide is a loop diuretic that primarily acts as an inhibitor of the sodium–potassium–chloride cotransporter in the tubular epithelium of the thick ascending loop of Henle. Furosemide uncouples sodium reabsorption in the renal medulla [85]. Patients with renal failure benefit from loop diuretics, as they retain their usefulness despite low GFRs. In contrast, distal diuretics theoretically lose efficacy when GFRs are low [85]. A negative effect of this diuretic is increased sodium excretion, a similar effect observed with ARBs. Therefore, monitoring serum sodium levels is suggested [86]. In advanced CKD, metabolic acidosis appears because of hydrogen ion retention and limits the intratubular entry of loop diuretics. Correction of this acidosis can optimize the response to diuretics (this was a prospective, randomized, controlled clinical trial conducted by the Nephrology Department of Zhejiang Hospital in China between June 2016 and July 2019) (Table 1) [87].

### 3.4. Mineralocorticoid Receptor

The mineralocorticoid receptor (MR) belongs to the nuclear hormone receptor subfamily and is expressed in various tissues and cell types, including the kidney, heart, vasculature, immune cells, and fibroblasts. It is co-expressed with the enzyme 11β-hydroxysteroid dehydrogenase type 2 (11β-HSD2), which converts cortisol into cortisone. The MR plays a pivotal role in the regulation of fluid balance, electrolyte homeostasis, and blood pressure (BP) [128]. The presence of 11β-HSD2 “safeguards” the MR by converting cortisol into its inactive form, cortisone [129]. During CKD and heart failure (HF), MR is often subject to overactivation. Beyond its role in increasing salt and water retention, MR overactivity amplifies the expression of target genes associated with inflammatory and fibrotic pathways, ultimately leading to organ damage. Blocking MR represents an appealing pharmacological strategy to safeguard organ function. This is particularly crucial in individuals with T2DM, as early stage kidney injury and CKD are linked to a higher likelihood of cardiovascular morbidity compared to progressing to end-stage renal disease (ESRD). The well-established beneficial effect of MR antagonists (MRAs) in reducing mortality among HF patients is widely recognized. Clinical studies have also provided evidence of the advantages of MRAs for individuals with CKD (FIDELITY amalgamates individual patient data from FIDELIO-DKD NCT02540993 and FIGARO-DKD NCT02545049) [130]. Few preclinical findings have made their way into therapeutic applications as swiftly and effectively as MRAs. These clinical trials have further underscored the efficacy of MR antagonists in reducing proteinuria and curtailing cardiovascular events in human CKD [131,132]. Despite the reno-protective benefits, MRAs have not garnered significant attention as viable agents for treating T2DM, primarily because of concerns about hyperkalemia and acute kidney injury. In addressing the risk of acute kidney injury, finerenone, a potent, selective, and nonsteroidal MRA (NSMRA), has emerged as a promising agent for reducing albuminuria and impeding the progression of acute kidney injury to CKD. Additionally, clinical trials within the FIDELITY program have shown that finerenone effectively reduces albuminuria and delays the decline in the glomerular filtration rate (GFR) in CKD patients (refer to Table 1) [131].

### 3.5. Hypoglycemic Agents

#### 3.5.1. Insulin Sensitizers

Insulin sensitizers are a class of hypoglycemic agents that aim to enhance insulin action. They achieve this by either increasing insulin secretion, improving peripheral insulin response, or enhancing glycemic control [90]. Among these, biguanides, such as metformin, are notable for their ability to reduce hepatic neoglucogenesis, lower fasting blood glucose levels, and ameliorate peripheral insulin resistance [90]. Given the substantial prevalence of metformin use and the heightened risk of associated lactic acidosis in CKD patients, it is considered best practice to refrain from prescribing metformin to individuals with an eGFR < 45 mL/min/1.73 m^2^ or those at high risk of acute kidney injury (AKI), irrespective of their baseline eGFR [89]. Second-generation sulfonylureas like glibenclamide, gliclazide, and glimepiride operate by lowering glucose levels through the stimulation of insulin secretion from pancreatic cells. This increased secretion corresponds to elevated circulating insulin levels. Prolonged use of these agents is associated with a progressive reduction in beta cell insulin production, a well-recognized contributor to secondary renal failure (refer to Table 1) [90,91].

#### 3.5.2. Dipeptidylpeptidase-4 (DPP-4i) Enzyme Inhibitors

Physiologically, dietary intake triggers the production of intestinal peptides that enhance insulin secretion, inhibit glucagon secretion, and reduce postprandial glucose levels. This phenomenon is referred to as the “incretin effect”. The primary incretins involved are glucagon-like peptide 1 (GLP-1) and glucose-dependent insulinotropic peptide (GIP). Dipeptidyl peptidase-4 (DPP-4) rapidly degrades these peptides, diminishing their insulinotropic impact. DPP-4 belongs to the family of cell membrane proteins and is expressed in various tissues, including immune system cells [90,133]. Dipeptidyl peptidase-4 inhibitors are a novel class of oral antidiabetic medications employed for the management of T2DM. They significantly lower glycosylated hemoglobin levels, pose a minimal risk of hypoglycemia, and cause no weight gain. For individuals with T2DM and compromised renal function, maintaining adequate glycemic control is imperative to delay the progression of renal dysfunction. Some gliptins, like vildagliptin, can be safely administered to patients with varying degrees of renal impairment, necessitating dose adjustments based on renal function [92]. Particular attention should be paid to the effects of linagliptin on impeding the progression of renal damage in individuals with DM and CKD during the nonreplacement phase, as well as its impact on proteinuria and glycemic control. These findings highlight the potential of linagliptin as a novel therapeutic approach owing to its renoprotective properties (conducted at the Nephrology and Internal Medicine Outpatient Unit at Kayseri Tertiary Care Research Hospital).

#### 3.5.3. Sodium–Glucose Cotransporter 2 Inhibitor (SGLT2i)

SGLT2 transporters encompass a wide range of membrane proteins responsible for transporting glucose, amino acids, vitamins, ions, and osmolytes across the brush border of the proximal renal tubules and the intestinal epithelium [134]. Increasing evidence supports the role of SGLT2i therapy in patients with CKD, including data from trials involving patients both with and without T2DM. Furthermore, studies have demonstrated that canagliflozin or dapagliflozin significantly reduce the risk of CKD progression. Notably, these trials marked the first instances where a reduction in all-cause CKD-related mortality risk was observed. Based on these findings, tailored treatment with SGLT2i emerges as a promising therapeutic avenue, capable of delaying disease progression in patients with CKD, regardless of diabetic status. In a randomized study involving placebo and canagliflozin treatment in patients with T2DM, substantial evidence showed a marked reduction in CKD progression risk, as indicated by decreased creatinine levels, reduced albuminuria and proteinuria, and improved serum hyperglycemia (DAPA-CKD, NCT03036150, conducted from 2 February 2017 to 12 June 2020) [94,95,96,97]. Research on the application of SGLT2 as specific reno-protective agents continues to advance. For the past two decades, RAS inhibitors have represented the sole therapeutic option for CKD patients. However, a recent study involving dapagliflozin in nondiabetic CKD patients has unveiled its potential for providing cardiorenal protection. Additional evidence has underscored the reno-protective effects of dapagliflozin in nondiabetic CKD patients. Consequently, the findings concerning dapagliflozin carry substantial implications for all CKD patients (DAPA-CKD, NCT03036150, conducted from 2 February 2017 to 12 June 2020) [135]. Clinical and preclinical investigations have elucidated multiple mechanisms responsible for SGLT2 inhibition. The advantages of such inhibition encompass enhanced cardiac and renal energy efficiency, cardiac remodeling, preservation of renal function, immunomodulation, hematocrit alterations, and control of CKD risk factors. Consequently, SGLT2is hold the potential to enhance outcomes for patients grappling with heart failure and CKD (refer to Table 1) [136].

### 3.6. Diet

Diet plays a crucial role and warrants consideration in patients with kidney disease. Dietary control can serve as a renal protective measure during the early stages of dialysis treatment, preventing malnutrition and obesity. It also holds the potential to delay the initiation of dialysis [137] and improve several metabolic markers. The specific dietary regimen employed depends on the stage of CKD, its progression, and the medications prescribed. Generally, the diet for renal patients is restrictive, though it does not compromise nutritional adequacy. Significant reductions in protein, phosphorus, calcium, and potassium intake can render diets less wholesome, and in some cases, more detrimental than beneficial [138,139]. Nutritional issues often manifest even before the onset of moderate renal failure [98], becoming increasingly significant in the predialysis phase [99]. Consequently, patients’ nutritional statuses necessitate close monitoring, with nutrient deficiencies addressed without impairing renal function. In most instances, very low-protein diets (VLPD) are recommended. Therefore, supplementation with ketoanalogs, such as ketoskeril, becomes essential for predialysis patients [100]. The adoption of VLPDs in CKD (stage 4) (GFR < 15 mL/min/1.73 m^2^) promotes delaying the initiation of dialysis [140]. Foods with a high biological value, such as eggs, have been proposed for individuals with renal disease (RD), given their richness in leucine, lutein, zeaxanthin, and vitamin D. However, the consumption of high biological-value foods should be regulated in accordance with the progression stage of renal disease [141]. The quantity and portion of egg consumption should also be considered. Nutritional counseling during CKD should not solely focus on food quantity and quality but also consider patients’ environmental and social behaviors [98]. Patients with CKD must demonstrate a willingness to modify their diets to enhance health-related outcomes, as age factors (≥80 years) may introduce complications associated with a reduced inclination to embrace new dietary options [142]. In CKD, diets featuring a predominance of vegetables (e.g., the Mediterranean diet) are recommended. These diets are rich in fiber, low in protein, and exhibit potential in reducing dyslipidaemia, guarding against lipid peroxidation and inflammation, elevating adiponectin concentrations, and enhancing intestinal microbiota [98,143]. Moreover, they supply ample vitamin K and impart a low acid load to the body. As a result, these diets may play a role in improving aspects beneficial to RD patients, including vascular calcification and bone health, protein metabolism in the colonic environment, and circulating levels of uremic toxins originating from microorganisms [144,145]. A highly alkaline diet contributes to a decreased acid load, subsequently reducing the stimulation of aldosterone, angiotensin II, and endothelin, all of which are implicated in dietary acid excretion [146]. In regions like Eastern countries, where rice is a dietary staple, protein-free rice varieties with low potassium and phosphorus content prove valuable in the dietary therapy of CKD patients. Freshly processed, low-protein brown rice, while maintaining similar energy content to white rice, offers reduced levels of potassium and phosphorus alongside heightened dietary fiber, γ-oryzanol, and antioxidant activity. Furthermore, dietary fiber and γ-oryzanol play pivotal roles in stabilizing intestinal microbiota, ameliorating uremic dysbiosis, and addressing leaky gut syndrome. These characteristics underscore the potential health benefits of low-protein rice diets in slowing CKD progression [101,102]. Potassium, renowned for its antihypertensive properties that promote sodium excretion and the elimination of most ingested potassium through the kidneys, experiences declining serum levels in tandem with diminishing renal function. Hyperkalemia or hypokalemia can manifest in CKD patients and are associated with cardiac arrhythmias and sudden death. Consequently, potassium intake should be tailored according to the patient’s CKD stage [105]. Conversely, sodium intake in CKD should be individualized. In broad terms, the advantages of a well-balanced diet include the reduction in nitrogenous compounds, diminished ammonia and uremic toxin production, and a decrease in metabolic by-products of bacterial origin (e.g., trimethylamine [TMA] and TMA N-oxide) (see Table 1).

### 3.7. Treatment with Vitamins and Minerals

#### 3.7.1. Potassium

Hyperkalemia is a metabolic disturbance that frequently manifests in patients with chronic CKD. Medications targeting the cardiovascular system are associated with an elevation in potassium levels. ACE inhibitors and ARBs have been linked to hyperkalemia. Current guidelines recommend baseline laboratory assessments and periodic monitoring upon initiation of treatment. For patients with systolic blood pressures < 120 mmHg, potassium levels > 4.5 mmol/L, or estimated glomerular filtration rate (eGFR) < 60 mL/min/1.73 m^2^, it is advised to conduct potassium level monitoring within the initial 4 weeks, following the clinical practice guidelines on hypertension and antihypertensive agents in CKD (K/DOQI). Inadequate monitoring increases the risk of mortality, with notable clinical alterations and a rapid decline in renal function [103]. Low potassium levels, or hypokalemia, are associated with a swift deterioration in renal function because of diuretics, ARBs, ACE inhibitors, and malnutrition. These factors have detrimental effects on renal outcomes (Kaohsiung for Delaying Dialysis (ICKD) study ((KMUH-IRB-990198)), 11 November 2002, and 31 July 2010) [104]. Potassium exhibits an antihypertensive effect by promoting the excretion of sodium and the majority of ingested potassium through the kidneys. A decline in renal function is correlated with an increase in serum potassium levels (Table 1) [105].

#### 3.7.2. Iron

Iron deficiency anemia (IDA) is a common consequence of CKD. Inflammation is closely associated with CKD, which leads to increased ferritin and hepcidin levels, thereby inhibiting iron absorption and efflux and reducing iron availability for erythropoiesis [106]. Conversely, iron is an essential substrate for improving cardiac function in heart failure and CKD. Consequently, the administration of intravenous iron may confer beneficial effects on the heart, potentially reducing cardiovascular events (Pivotal Trial Number—No: 2014-004133-16 REC no: 14/YH/1209, 17 February 2015–23 May 2015) [107]. The efficacy of intravenous iron supplementation in IDA has been extensively studied in patients with various conditions, including postpartum anemia and gastrointestinal disorders. The use of ferumoxytol to treat IDA has been shown to increase hemoglobin (Hb) production in patients with CKD and those undergoing dialysis (CKD-201 and IDA-302, NCT01052779 15 January 2010 and NCT01114204, 29 April 2010) [108]. An initial iron dose of 500–1000 mg significantly improves CKD (FIND-CKD, NCT00994318, 13 October 2009–9 May 2014) [109]. However, iron therapy is associated with hypersensitivity reactions, hypophosphatemia, oxidative stress, and exacerbating infections. Iron isomaltoside has a limited association with severe hypersensitivity reactions (Clinical-Trials.gov (NCT01102413) from 26 March 2010 to 30 June 2010 to 25 April 2014) [110,111]. Nevertheless, these reactions occur in a small percentage of the population and are considered rare; therefore, intravenous iron administration can be considered safe. Patients should be closely monitored during treatment [112]. Additionally, ferric citrate increases Hb and iron levels while lowering serum phosphate and fibroblast growth factor 23 (FGF23) [113]. Limited studies have shown that early anemia treatment extends renal survival in CKD patients, including those not undergoing dialysis. However, initiating treatment in these patients when Hb levels fall below 11 g/dL but not below 10 g/dL can effectively reduce the risk of renal events (this study is registered with the University Hospital Medical Information Network (ID: UMIN000003116, February 2010 to March 2011)) [147]. Therefore, Hb levels should be raised to a minimum of 12 g/dL [148,149]. Erythropoiesis-stimulating agents (ESAs) are currently available and include short-acting recombinant human erythropoietin (epoetin), medium-acting darbepoetin alfa, long-acting epoetin beta pegol, and their biosimilars. These agents share a common mechanism of action but possess different pharmacokinetic and pharmacodynamic properties. Despite the widespread use of ESAs in treating anemia in CKD patients, the relative mortality risks associated with different ESAs are yet to be fully elucidated. However, long-acting ESAs may be linked to higher mortality risks in hemodialysis patients compared to those on short-acting ESAs [150]. Biosimilars have demonstrated safety and efficacy profiles comparable to those of the original ESAs, depending on the severity of anemia and iron deficiency [151]. There are two categories of poor responses to ESAs. In the first category, patients fail to exhibit an increase in Hb concentration despite repeated increases in ESA dose. In the second category, there is a reduced response to treatment despite increased ESA doses. Both categories necessitate a careful, systematic approach. Other strategies for treating anemia in CKD patients are currently under investigation. These strategies include stabilizing hypoxia-inducible factors with prolyl-hydroxylase inhibitors and hepcidin modulation strategies. The effects of these scientific developments on managing anemia in CKD patients are yet to be determined (Table 1) [152].

#### 3.7.3. Vitamin D

Studies have demonstrated that vitamin D enhances physical performance and provides protection against histological and radiological indicators of hyperparathyroidism secondary to CKD [153]. Patients with CKD who do not require dialysis commonly exhibit a significant deficiency in 25-hydroxyvitamin D (25(OH)D). The relationship between 25(OH)D levels and metabolic syndromes in individuals with severe CKD and without diabetes remains unclear [154]. In contrast, elevated concentrations of FGF23 contribute to 25(OH)D deficiency in CKD by stimulating the 24-hydroxylation of this metabolite, resulting in its subsequent degradation [114]. Vitamin D therapy may yield antiproteinuric effects and potentially reduce the rate of renal function decline. Beyond its primary role in calcium homeostasis and bone mineralization, vitamin D may assist in the suppression of the renin–angiotensin system (RAS) and exert cardiovascular, metabolic, and direct immunomodulatory effects (see Table 1) [115].

### 3.8. Calcimimetics

Calcimimetics play a crucial role in the management of secondary hyperparathyroidism. CKD is characterized by abnormal calcium and phosphorus levels in both the bloodstream and tissues, which are closely associated with reduced survival rates and arterial stiffening, contributing to the development of heart disease. Therapeutic interventions such as dietary restrictions, phosphorus binders, and vitamin D compounds have been proposed to address these abnormal mineral levels. Among these approaches, cinacalcet, a novel treatment, has shown promise in normalizing mineral levels, although its full effects on patients are still under investigation [116]. Studies evaluating the impact of cinacalcet in approximately 7500 individuals with CKD have demonstrated improvements in blood abnormalities and the effective control of parathyroid hormone secretion. Moreover, cinacalcet concurrently reduces calcium, phosphorus, and calcium–phosphorus products. These multifaceted benefits position calcimimetics as more advantageous than traditional therapies across the spectrum of secondary hyperparathyroidism severity (refer to Table 1 for details) [117].

## 4. Novel Proposals

### 4.1. Probiotic in CKD

The gut microbiome holds promise for the management of CKD, with Lactobacillus and Bifidobacterium emerging as the most commonly used probiotics. However, consensus regarding the appropriate dosage and duration of probiotic administration for CKD patients remains elusive. Recent evidence suggests that individuals in the early stages of CKD exhibit an altered microbiota profile, although controlled studies investigating probiotic usage in this population are lacking. Advanced CKD has been associated with disturbances in the gut microbiota. Positive outcomes have been reported in randomized controlled trials employing probiotics for nondialysis CKD stages 3–5; however, studies involving the dialysis population have yielded mixed results [155]. The ProLowCKD trial (registration number NCT04204005, conducted from 13 March 2017 to 31 December 2020) demonstrated that this treatment effectively controlled and modulated microbiota-derived proatherogenic toxins in CKD patients [156]. In the CKD-REIN clinical trial (NCT03381950, conducted from 10 November 2010 to 11 October 2019), the consumption of yogurt and probiotics was associated with a reduced risk of inflammation in CKD patients [157]. The INCMNSZ clinical trial of 2014 (conducted at the Instituto Nacional de Ciencias Médicas y Nutrición Salvador Zubirán, Mexico) focused on a simple randomized controlled study examining changes in blood urea levels. The results demonstrated a reduction of more than 10% in serum urea concentrations [158]. The Observational Study of Kibow Biotics in Chronic Kidney Failure Patients on Dialysis (NCT01450709, conducted from April 2011 to November 2012) observed biochemical parameters such as blood urea nitrogen (BUN), serum creatinine, and uric acid. The results revealed statistically significant differences in BUN levels between the placebo and probiotic treatment groups [159]. The NATURE 3.1_New Approach for the Reduction of REnal Uremic Toxins (NATURE31) clinical trial (NCT03815786, conducted from 24 January 2019 to 15 February 2022) demonstrated the effectiveness of the symbiotic NATUREN G^®^ in reducing serum-free, mild intestinal permeability, abdominal pain, and constipation syndromes [160].

### 4.2. Metabolic Modulators

Peroxisome proliferator-activated receptor gamma (PPAR-γ) is one of three nuclear PPAR receptors that function as ligand-activated transcription factors. PPAR-γ regulates lipid, glucose, and amino acid metabolism in immune cells, the skin, and various organs. The receptor translates nutritional, pharmacological, and metabolic stimuli into alterations in gene expression. PPAR-γ activation promotes cell differentiation, reduces proliferation, and modulates the immune response. PPAR-γ is expressed in all glomerular, tubular, and tubulointerstitial cell types, playing a pivotal role in the pathogenesis of various renal diseases. Over the past decade, accumulating evidence based on systemic PPAR-γ activation has demonstrated an antifibrotic effect by regulating the action of TGF-β1, the principal mediator of fibrogenesis in most organs. Conversely, TGF-β1 downregulates PPAR-γ in a murine model of CKD (see Table 2) [11,161].

### 4.3. Novel Antihypertensives

#### 4.3.1. Vasopressin Inhibitors

Tolvaptan, a vasopressin V2 receptor blocker, demonstrates diuretic effects in heart failure and the impact on patients with CKD. V2 receptors for arginine vasopressin are located in the renal collecting duct. These receptors promote water reabsorption through aquaporin-2. Studies have revealed elevated vasopressin concentrations in patients with advanced CKD; consequently, the kidney is unable to produce sufficient urine. This excess volume results in systemic fluid overload, as evidenced by generalized edema [168]. The aquaretic effects of tolvaptan were confirmed in 21 CKD patients, where there was an increase in urine volume, a decrease in urine osmolality, and an increase in serum sodium concentration, effectively correcting hyponatremia. Loop diuretics are conventionally used in patients with renal impairment but are associated with adverse effects, such as hypokalemia and hyponatremia [169]. In contrast, tolvaptan facilitates water excretion without altering renal hemodynamics or sodium and potassium excretion; however, the effects of tolvaptan on urine volume and urinary serum sodium excretion in advanced CKD remain to be fully elucidated. In a double-blind phase III clinical study, the efficacy of tolvaptan in autosomal polycystic kidney disease was predicated on reducing cysts by lowering cAMP levels compared to a placebo. This study involved adult patients with rapid cyst growth. Patients commenced tolvaptan at 45 + 15 mg (45 mg upon awakening and 15 mg 8 h later), progressing to a regimen of 60 + 30 mg and 90 + 30 mg if tolerated by the patient, with at least one week of separation between different dosage changes. Additionally, a group of patients received a placebo. This study found that the efficacy of tolvaptan depended on renal function and imaging parameters, without providing direct information on whether it delays the need for dialysis or extends survival. Tolvaptan is reported to reduce renal growth, delay the onset of end-stage renal disease, and decrease associated mortality (TEMPO 4:4 was an open-label extension trial; participation was voluntary for TEMPO 3:4 (clinical trial identifier: NCT00428948, 2006-002768-24)) (Table 2) [170,171].

#### 4.3.2. Treatments Based on Angiotensin 1–7 Inhibitor

Angiotensin 1–7 (Ang 1–7) is generated by ACE type 2 from angiotensin II and exerts an antagonistic effect on the activity of this angiotensin [200]. At the renal level, Ang (1–7) plays a protective role as it improves glomerular morphology and function, as evidenced by a reduction in proteinuria, metabolite excretion (creatinine and urea), and the decrease in fibrogenic markers, such as TGF-β, TIMP-1, and TIMP-2. Consequently, Ang (1–7) induces the reversal of glomerulosclerosis, counteracting the effects of Ang II. Building upon these findings, Choi et al. (2020) concluded that Ang (1–7) exhibits antifibrotic, anti-inflammatory, and antiapoptotic actions by restoring the altered RAS, as observed in animal models. Thus, Ang (1–7) has the potential to inhibit the progression of CKD (see Table 2) [177,200].

#### 4.3.3. Endothelin Inhibitors

The specific roles of endothelin in the genesis of AHT and its overexpression in CKD are well established. Substances developed to block endothelin (such as sitaxentan and darusentan) have not proven effective because of their adverse reactions, including increased extracellular volume. Therefore, loop diuretics are employed to manage this adverse reaction [173,174]. Sitaxentan and darusentan have undergone extensive study, with no significant differences in efficacy compared to other classes of antihypertensives identified. These drugs were evaluated in 27 patients with proteinuria and underlying CKD. An increase in endothelin 1 expression was observed in all patients following the blockade, and patient fluid retention was also assessed [173]. The use of a selective endothelin A receptor antagonist (ETA) in combination with RAS inhibitors has been shown to prevent proteinuria in CKD [174]. One of the latest endothelin antagonists proposed is atrasentan, an ETA receptor antagonist used to reduce proteinuria in a model of male Dahl salt-sensitive rats. These 6-week-old rats were administered water with or without varying doses of atrasentan under high-salt diet or normal diet conditions for 6 and 12 weeks. At the end of the 12th week, a moderate dose of atrasentan was found to significantly attenuate proteinuria and serum creatinine without reducing mean arterial pressure (MAP). Consequently, cardiac hypertrophy was prevented, and renal function improved (see Table 2) [174].

### 4.4. Antifibrotics

Renal interstitial fibrosis is characterized by the accumulation of extracellular matrix proteins, a common hallmark of CKD. Recent studies have revealed four new mediators of renal fibrosis: discoidin domain receptor 1, periostin, connexin 43, and cannabinoid receptor 1. In experimental models, gene suppression techniques, such as antisense sequences or specific blockers, were applied, resulting in the delay of kidney disease progression. Renal structure and function were preserved even after the initiation of inhibitor therapy [201]. Histone deacetylases (HDACs) play critical roles in multiple cellular signaling pathways, including those involving TGF-β, epidermal growth factor receptors, the signal transducers and transcription activator 3 pathway, and the JNK/Notch2 signaling pathway. These pathways influence inflammation, oxidative stress, and angiogenesis, which are pivotal processes in fibrosis development. Therefore, HDAC inhibitors could represent a promising therapeutic strategy [202]. Similarly, the Wnt/β-catenin signaling pathway has been associated with the development and progression of renal fibrosis. Thus, inhibiting this signaling pathway may yield benefits and can be achieved through various antagonists, such as frizzled-related proteins (FZD), the Dickkopf 1 protein family, Klotho, and Wnt-1 inhibitory factor [203]. Studies conducted on animal models with diabetic kidney disease have identified selonsertib as a selective inhibitor of apoptosis signal-regulating kinase. This discovery suggests that selonsertib holds potential as a therapeutic agent [178]. The G-protein-coupled CXC chemokine receptor 4 (CXCR4) emerges as a potential therapeutic target for tissue fibrosis. In that study, the impact of CXCR4 inhibition on the expression of extracellular matrix (ECM) components, matrix metalloproteinase-2 (MMP-2), and several signaling pathways related to transforming growth factor β1 (TGF-β1) was assessed. The results demonstrated that blocking CXCR4 reduced the activity of the p38 MAPK-Kinase, PI3K/AKT/mTOR, and STAT3 signaling pathways implicated in renal fibrosis. These findings suggest that CXCR4 inhibitors could represent a novel strategy to curtail the progression of renal fibrosis (see Table 2) [179].

### 4.5. Antioxidants

Patients with CKD have a high incidence of cardiovascular disease and experience elevated cardiovascular morbidity and mortality because of endothelial dysfunction and left ventricular hypertrophy, which can be triggered by oxidative stress and inflammation. The structural features of CKD, renal energy loss, and uremic toxins (urea, creatinine, and uric acid) lead to an imbalance between free radical production and antioxidant defenses. Additionally, patients with CKD often present with multiple cardiovascular risk factors, such as diabetes mellitus, dyslipidemia, and hypertension, which are associated with oxidative stress resulting from reactive oxygen species (ROS). ROS can trigger the inflammatory process and accelerate the progression of renal injury. ROS are generated through several major enzymatic processes, such as nicotinamide adenine dinucleotide phosphate (NADPH) oxidase, which reduces oxygen to a superoxide anion (O_2_^−^). This anion is subsequently converted to hydrogen peroxide (H_2_O_2_) by superoxide dismutase (SOD). The superoxide anion (O_2_^−^) reacts with nitric oxide, resulting in the production of peroxynitrite (nitrosative stress). Hydrogen peroxide reacts with intracellular iron to form hydroxyl radicals. Additionally, in the presence of chloride ions, H_2_O_2_ is catalyzed to hypochlorous acid by myeloperoxidase activity. Excessive ROS can lead to the oxidation of lipids, proteins, and DNA [204,205]. In CKD patients, inflammatory processes and oxidative stress predominate, including increased TNF-α, IL-6, and NF-κB activation, which also alter the immune system and adipose tissue through the production and secretion of adipokines, contributing to a systemic inflammatory state in CKD. Some biomarkers have proven suitable for the CKD population, such as malondialdehyde, lipid hydroperoxides, F2-isoprostanes, asymmetric dimethylarginine (ADMA), protein carbonyls, AOPPs, 8-oxo-7,8-dihydro-2’-deoxyguanosine, and glutathione-related activity [204]. Several natural compounds can reduce the inflammatory state by modulating the NF-κB-dependent pathway, reducing the levels of proinflammatory cytokines and ROS and increasing the levels of antioxidant enzymes. Fat-soluble antioxidants present in the cell membrane include vitamin E, β-carotene, and coenzyme Q. Water-soluble antioxidants include vitamin C, glutathione peroxidase (glutathione-Px), SOD, and catalase. Ferritin, transferrin, and albumin exert a nonenzymatic antioxidant effect by sequestering the transition of metal ions. The primary enzymatic antioxidant defense is SOD, which accelerates the dismutation rate of oxygen to H_2_O_2_, while catalase reduces H_2_O_2_ to water. Glutathione peroxidase reduces H_2_O_2_ and other organic peroxides to water and oxygen, requiring glutathione as a hydrogen donor—a scavenger for H_2_O_2_, hydroxyl radicals, and chlorinated oxidants. Decreased levels of glutathione and plasma GP activity have been reported in CKD patients [204,205]. On the other hand, various antioxidants, including vitamin E, vitamin C, curcumin, resveratrol, green tea, and other metabolites, flavonoids, and polyphenols derived from plant species [206], can ameliorate oxidative stress in CKD patients. Vitamin E protects the cell membrane from lipid peroxidation by forming a low-reactivity tocopheroxyl radical. Vitamin C directly scavenges O_2_- and hydroxyl radicals. Supplementation with catechin (a type of natural phenol and antioxidant), vitamin E, and vitamin C has been shown to decrease malondialdehyde and ADMA levels in kidney failure patients, in addition to its beneficial effect on controlling hypertriglyceridemia. Administration of omega-3 fatty acids ameliorates oxidative stress, as evidenced by the reduction in thiobarbituric acid-reactive substances (TBARS) and the increase in serum SOD, GP, and catalase activities in CKD patients with dyslipidemia. N-acetylcysteine administered to children undergoing dialysis has been found to reduce intracellular oxidative stress in T cells and can effectively lower the oxidative stress-induced functional disability of lymphocytes. Similarly, coenzyme Q10 administration has been shown to suppress both oxidative stress and antioxidant indices, partially reducing the oxidative state. Folic acid counteracts lipoperoxidation and hyperhomocysteinemia in patients on hemodialysis and can help decrease cardiovascular risk in this population. Treatment of CKD patients with the lipid-lowering agent atorvastatin has been shown to reduce oxidized LDL-C, total cholesterol, triglycerides, LDL-C, and apolipoprotein B in patients on dialysis. Selenium is an essential trace element with antioxidant properties, which is incorporated into GP as selenocysteine and plays an important role in cellular protection as a free radical scavenger. Finally, zinc, green tea extract, and silymarin are potential antioxidants for CKD patients [204,205]. Icariin, a flavonoid from the Epimedium plant, restores the damaged renal histological structure and decreases urea nitrogen, creatinine, and uric acid levels. It also reduces TGF-β protein and gene expression and induces the proliferation of renal stem cells CD133 [207]. Another flavonoid, quercetin, inhibits the fibrogenic process induced by TGF-β1 [180]. Other sources that induce an antioxidant response include natural products that stimulate the Nrf2 response. This response is a central mechanism for the nuclear antioxidant response (ARE) and the synthesis of antioxidant enzymes. Methyl bardoxolone binds and inactivates KEAP1, allowing Nrf2 to act in the nucleus and activate the ARE response. Another natural source of antioxidant compounds is broccoli seeds, which contain large amounts of glucoraphanin. Glucoraphanins are processed in the stomach and release sulforaphane (SFN), which induces significant increases in the activities of the phase II detoxification enzymes NADPH dehydrogenase, quinone 1, and glutathione transferase (activation products of Nrf2) in the stomach, small intestine, and liver of wild-type mice, but not in Nrf2 KO mice. Propolis, a natural product rich in prenylated flavonoids, exerts nephroprotective effects through the regression of renal fibrosis [181]. Shi et al. [186] proposed the Bushen Huoxue compound (BSHX), which includes active principles of natural compounds such as tanshinone IIA, rhein, curcumin, calycosin, and quercetin from traditional Chinese medicine (TCM). These compounds act on processes such as inhibiting TGF-β and signaling pathways (Table 2) [186].

### 4.6. Retinoic Acid (RA) (Vitamin A)

Retinoids derived from vitamin A play a pivotal role in renal development, influencing tubulogenesis and nephron count [208]. Retinoids have been demonstrated to confer protection against renal injury in various experimental kidney disease models [209]. In models of acute and chronic mesangio proliferative glomerulonephritis, retinoids preserve renal function, reduce albuminuria, and mitigate glomerular and tubular damage. The protective effects of retinoids have also been documented in mouse models of diabetic nephropathy [210] and in a model simulating antibody-mediated podocyte injury [211]. However, it is imperative to consider the potential side effects of retinoic acid (Table 2).

### 4.7. Oral Carbon Absorbent (AST-120)

AST-120 (Kremezin) represents an activated, carbonated oral absorbent widely utilized in Oriental countries. It possesses an adsorption capacity for specific low-molecular-weight organic compounds known to accumulate in patients with CKD. Oral administration of AST-120 in uremic rats and patients with CKD has been observed to reduce elevated serum indoxyl sulfate levels. In a study involving patients with CKD in Japan, AST-120 suppressed the increase in serum creatinine levels, prevented proteinuria, improved uremic symptoms, and resulted in the postponement of dialysis therapy. The effect of AST-120 in patients with late-stage CKD remains uncertain. Nevertheless, a meta-analysis incorporating 15 randomized controlled trials reported, with a confidence index of 95%, that the personalized dosage of AST-120, when compared with another administered dosage, represented the optimal treatment goal for patients with CKD [188]. Conversely, the multinational, randomized, double-blind, placebo-controlled Evaluating Prevention of Progression in CKD trials (EPPIC-1 and EPPIC-2 trials, 07/2007–02/2012) assessed the impact of AST-120 on CKD progression when added to standard therapy. The results revealed that disease progression occurred more gradually than anticipated in the trial populations. The data from these trials did not support the benefit of adding AST-120 to standard therapy in patients with moderate to severe CKD [189]. On the other hand, AST-120 has been found to attenuate AKI-induced cardiac injury and enhance cardiac function in mice, potentially mitigating the development of cardiorenal-syndrome-related CKD. This discovery has been attributed to the drug’s capacity to suppress apoptosis and proinflammatory NF-κB/ICAM-1 signaling (Table 2) [190].

### 4.8. Glucagon-like Peptide-1 Receptor Agonist (GLP-1RA)

The GLP-1 receptor is expressed in humans in the pancreas, lungs, brain, kidneys, stomach, and heart but not in the liver, skeletal muscle, or adipose tissue. Binding between GLP-1 and its receptor activates adenylate cyclase, leading to an immediate increase in cyclic AMP and cytoplasmic Ca+2, resulting in insulin secretion. In addition to the short-term effect of GLP-1 on enhancing glucose-dependent insulin secretion stimulation, continuous GLP-1 activation also increases insulin synthesis, modulates β-cell proliferation, and inhibits β-cell apoptosis and glucagon release. Incretin hormones also reduce gastric emptying, suppress food intake, and enhance natriuresis and diuresis [212].

GLP-1RA agonists are antihyperglycemic drugs that enhance cardiovascular and renal health in diabetic nephropathy. GLP-1RA offers the potential for adequate glycemic control in multiple stages of diabetic nephropathy without an increased risk of hypoglycemia. It also prevents the onset of macroalbuminuria and slows the decline in GFR in patients with diabetes. Therefore, GLP-1RA provides the additional benefit of regulating blood glucose, reducing body weight by inhibiting food intake and reducing gastric motility, stimulating cell proliferation, reducing inflammation and apoptosis, and improving cardiovascular function, neuroprotection, and renoprotection. Results from ongoing trials are pending to assess the impact of GLP-1RA treatments on primary renal endpoints in diabetic nephropathy [191,192,193,194].

Based on the results of clinical trials, the current guidelines of the American Diabetes Association and Korean Diabetes Association recommend that clinicians consider prescribing SGLT2 inhibitors or GLP-1 receptor agonists after metformin as part of the glucose-lowering regimen for patients with T2DM and CKD [212]. Less than a quarter of patients treated with GLP-1RAs (22.4%) had a beneficial metabolic response according to the NICE and CatSalut 6-month continuation criteria (HbA1c reduction ≥ 1% with weight loss ≥ 3%), which is consistent with the 25–34% reported by some other studies conducted in primary care. This finding would argue for an independent assessment of weight and glycemia reduction when deciding to continue treatment with GLP-1RAs, further supported by the results of the multivariate analysis. The prescription of these drugs represents noncompliance with the prescription instructions, which recommend not using them in patients with terminal or severe CKD and using them with caution—or adjusting the dose—in moderate CKD. Thus, in a study by Franch-Nadal et al. (2019), the mean eGFR decreased by 1.63 and 1.2 units at 6 and 12 months, respectively. However, evidence from available studies suggests that patients with T2DM and CKD are less likely to develop or experience greater deterioration in renal function when treated with GLP-1RA than with placebo or other hypoglycemic agents (refer to Table 2) [213].

### 4.9. Demethylating Agents

RNA methylation is a key process in the epigenetic regulation of post-transcriptional gene expression. Abnormal methylation is implicated in various renal diseases [214]. For example, RNA methylation (m6A) plays an important role in AKD and CKD, as well as in their progression from AKD to CKD. Elevated levels of YAP and YTHDF1 methylation enzymes have been observed, implicating m6A methylation enzymes (METTL3, METTL14, and WTAP) in the diagnosis, treatment, and future therapeutic opportunities for kidney diseases [215]. The known pathogenic mechanisms after acute kidney injury (AKI) to CKD transition include aberrant repair, cell cycle arrest (G2/M) of tubular epithelial cells, perpetuated fibroblast activation, microvascular rarefaction, chronic inflammation, and sustained activation of the RAS after AKI. Consequently, residual focal fibrosis and activated myofibroblasts in kidneys with partial AKD recovery are pathological findings possibly related to the subsequent transition to CKD. In a model of folic-acid-induced progressive renal fibrosis, Bechtel et al. [216] demonstrated that hypermethylation of Rasal1, which encodes an inhibitor of the Ras oncoprotein, leads to perpetuated activation of myofibroblasts, contributing to renal fibrosis development. On the other hand, hypermethylation of the erythropoietin promoter and enhancer in myofibroblasts has also been shown to lead to decreased erythropoietin production and anemia in CKD [198]. Currently, two demethylating agents are in clinical use, 5-azacytidine and decitabine (5-aza-20-deoxycytidine), which are approved for the treatment of specific forms of myelodysplastic syndrome and acute myeloid leukemia. However, these drugs have also demonstrated potential roles in alleviating renal fibrosis [197]. Demethylation by 5-azacitidine (5-Aza) not only increases erythropoietin expression but also reduces Acta2 expression, suggesting the potential effect of demethylation in reverting myofibroblasts to pericytes (refer to Table 2) [197,198].

### 4.10. Novel Potassium Binders

Two novel potassium binders, sodium zirconium cyclosilicate and patiromer, have been employed in patients experiencing hyperkalemia because of heart failure or CKD. These drugs are akin to existing potassium binders and exchange dietary potassium for sodium or calcium in the intestine, thereby preventing potassium absorption. Both drugs were assessed against a placebo in patients with CKD who were developing hyperkalemia while receiving RAAS inhibitors. These drugs effectively reduced serum potassium levels and were reasonably well tolerated even when co-administered with RAAS inhibitors. Consequently, patients at high cardiovascular risk may derive benefits from the protective effects of RAAS inhibitors. Limitations encompass a relatively brief treatment period, the absence of a control group employing existing potassium binders, and the exclusion of patients with severe or symptomatic hyperkalemia [217,218].

### 4.11. Hypoxia-Inducible Factor (HIF) Stabilizers

A novel strategy for treating anemia involves the utilization of protein inhibitors known as HIFs. This approach entails the application of a protein to inhibit the degradation of HIFs by the proteasome. Consequently, the erythropoietin gene is activated, leading to the endogenous production of erythropoietin [219]. Clinical trials assessing the use of HIFs for mitigating anemia in patients with improved safety and efficacy are currently underway (HIF-PHI program, 2019 to the present) [220]. Prolyl hydroxylase inhibitors (PHI) of hypoxia-inducible factor (HIF) have demonstrated efficacy in treating anemia in patients with chronic kidney disease (CKD). These agents function by stabilizing the HIF complex and promoting the endogenous production of erythropoietin, even in patients with end-stage kidney disease. This approach is especially relevant since the use of recombinant human erythropoietin and its derivatives has been associated with a potential increased risk of myocardial infarction and other adverse events [221,222].

### 4.12. Stem Cells of Mesenchymal and Adipose Tissue Origins

In the realm of CKD treatment, traditional approaches do not hold the promise of a cure or the ability to halt disease progression. Consequently, innovative strategies have surfaced as potential therapeutic options for safeguarding renal function. These emerging approaches are centered on regenerative cell-based therapies, celebrated for their multifaceted impacts, including paracrine signaling, immunomodulation, regulation of inflammation, secretion of various trophic factors, and the intriguing potential for differentiation into renal precursors [162]. Endothelial-derived progenitor cells and mesenchymal stromal cells (MSCs) derived from adipose tissue contribute significantly to neovasculogenesis. Their primary mechanism involves paracrine-driven angiogenesis and their ability to transform into fully functional endothelial cells within the kidney’s microenvironment. Notably, MSCs exert paracrine influences, effectively promoting proangiogenic, anti-inflammatory, and antifibrotic activities [223]. Experimental studies utilizing murine models of nephrectomy have demonstrated that MSC transplantation can attenuate interstitial fibrosis and slow the progression of CKD through these paracrine effects. Furthermore, MSCs play a pivotal role in safeguarding against epithelial–mesenchymal transition, effectively counteracting TGF-β1-induced loss of podocyte synaptopodin [224]. In clinical practice, the intravenous administration of MSCs at a dosage of 1 × 106 cells/kg to CKD patients has shown promise. This treatment approach results in improved urinary protein excretion without compromising the glomerular filtration rate (GFR) or eliciting any adverse effects. It is important to underscore the necessity for vigilant patient monitoring following this intervention to assess potential risk factors for cancer development and the emergence of anti-human leukocyte antigen antibodies [225]. Although certain studies have provided insights into how stem cells orchestrate the regulation of distinct signaling pathways linked to renal fibrosis, including TGF-β/Smad, NF-κB, MAPK/ERK, phosphatidylinositol-3 kinase/protein kinase B (PI3K/AKT), and TNF-α, our understanding of the in vivo mechanisms of action remains incomplete. Crucial questions concerning the optimal cell source, the most effective route of administration, and the precise dosage for treating various renal diseases persist, indicating the need for further exploration in this field [163] (Table 2).

### 4.13. Antagonism of the Sympathetic Nervous System or Sympathectomy

Activation of the sympathetic nervous system is a pivotal mechanism in the development of hypertension and its associated comorbidities. Postganglionic efferent sympathetic nerve fiber activation leads to increased renin release, tubular sodium reabsorption, and a reduction in renal blood flow [184,226].

#### Sympathectomy

The neurotransmitter norepinephrine in the renal sympathetic system acts on epithelial cells via the α2A-adrenoreceptor, promoting cellular senescence through β-arrestin2 signaling. This represents a potential target for preventing renal fibrosis. By inhibiting this pathway with receptor antagonists, it is possible to reduce senescence [195].

Sympathetic hyperactivity emerges early in chronic kidney disease (CKD) and intensifies significantly as CKD progresses to end-stage renal disease (ESRD). Effective blood pressure control is crucial for slowing CKD progression. Renal sympathetic denervation has recently emerged as a potent intervention for managing refractory hypertension [139]. Renal catheter denervation is a treatment modality specifically designed to counteract sympathetic hyperactivity, commonly observed in patients with uncontrolled hypertension. It leads to a reduction in muscle sympathetic nervous activity, renal norepinephrine spillover, and blood pressure in various cohorts of hypertensive patients [226].

Q. Li et al. [195] observed a significant increase in renal fibrosis and cellular senescence in models of unilateral ureteral obstruction and ischemia–reperfusion injury. Renal denervation partially reversed the effects of unilateral ureteral obstruction and unilateral ischemia–reperfusion injury. In vitro studies demonstrated that norepinephrine induced epithelial cells to secrete proinflammatory cytokines and promoted cellular senescence by activating α2A-AR (α2A-adrenergic receptor). The effects of norepinephrine during cellular senescence were mitigated by a selective antagonist of α2A-AR and β-arrestin2 siRNA (which acts downstream of α2A-AR). Notably, several immune system cells and cells associated with fibrotic processes express receptors for neurotransmitters. In contrast, those related to the sympathetic nervous system predominantly activate the inflammatory and fibrogenic processes (see Table 2) [227,228].

## 5. Benefit of the Treatment in Population with CKD

The incidence, prevalence, and progression of CKD exhibit significant variations among different countries, primarily because of ethnic factors and social determinants of health. In this context, the primary objective of the present study is to thoroughly examine the currently available therapies for CKD and identify therapeutic strategies that could contribute to mitigating the uncontrolled growth of the population affected by this condition. These innovative therapeutic options hold substantial potential for alleviating the burden CKD imposes on public health by significantly reducing the costs associated with procedures such as hemodialysis, peritoneal dialysis, and costly hospitalizations. Ultimately, this approach has the potential to alleviate the economic burden imposed by conventional CKD treatments on public healthcare systems.

### Future Direction

Chronic kidney disease (CKD) is a severe condition influenced by various etiological factors. Existing treatments have proven ineffective in halting CKD progression. Consequently, most advanced CKD patients develop complications necessitating dialysis or renal transplantation. At present, the focus remains on managing hypertension, water and electrolyte imbalances, and associated comorbidities like diabetes mellitus (DM), yielding modest results in CKD management. The imperative to devise innovative treatment approaches targeting CKD’s underlying processes is evident. Current strategies primarily revolve around the use of antifibrotics and antioxidants that inhibit specific receptors associated with proinflammatory and profibrotic signaling pathways, alongside reducing ROS levels. A novel discovery includes the alternative application of sympathetic system inhibitors (alpha or beta adrenoblockers) to maintain an antifibrotic state in mesangial cells. Undoubtedly, one of the most promising avenues for CKD treatment involves regenerative therapies, particularly stem cell therapy. Stem cells offer a positive outlook because of their attributed paracrine effects, immunomodulatory properties, regulation of inflammation, secretion of various trophic factors, and potential for renal precursor differentiation. Furthermore, another intriguing perspective in these treatments involves the activation of the PPAR-γ (gamma) receptor, responsible for lipid metabolism regulation and its association with promoting cell differentiation. In CKD models, PPAR-γ activation demonstrates an antifibrotic effect by regulating TGF-β1. Additionally, there has been consideration of compounds capable of adsorbing low-molecular-weight organic molecules that accumulate in CKD patients. The examination of these novel therapeutic options for CKD aims to identify combination therapies capable of reversing renal damage associated with the condition and the development of new drugs targeting molecular-level signaling pathways (such as TGF-β and Smad, Nrf2) (Figure 1).

## 6. Conclusions

Chronic kidney disease (CKD) has inflicted substantial morbidity and mortality on a global scale, and this trend continues to escalate. The prevailing treatments, unfortunately, prove to be ineffective in halting the progression of kidney dysfunction or ameliorating the complications arising throughout the course of the disease.

The realm of research and drug development, however, has introduced a ray of hope by exploring a diverse array of therapeutic agents, each targeting more specific facets of CKD pathology. These promising avenues hold the potential to enhance our ability to manage hypertension, regulate glycemia, and rectify electrolyte imbalances, some of which exhibit pronounced antifibrotic effects.

Moreover, these ongoing research endeavors raise the tantalizing prospect of devising combination therapies with the capacity to reverse the renal damage that plagues those afflicted by advanced CKD.

## Figures and Tables

**Figure 1 biomedicines-11-02828-f001:**
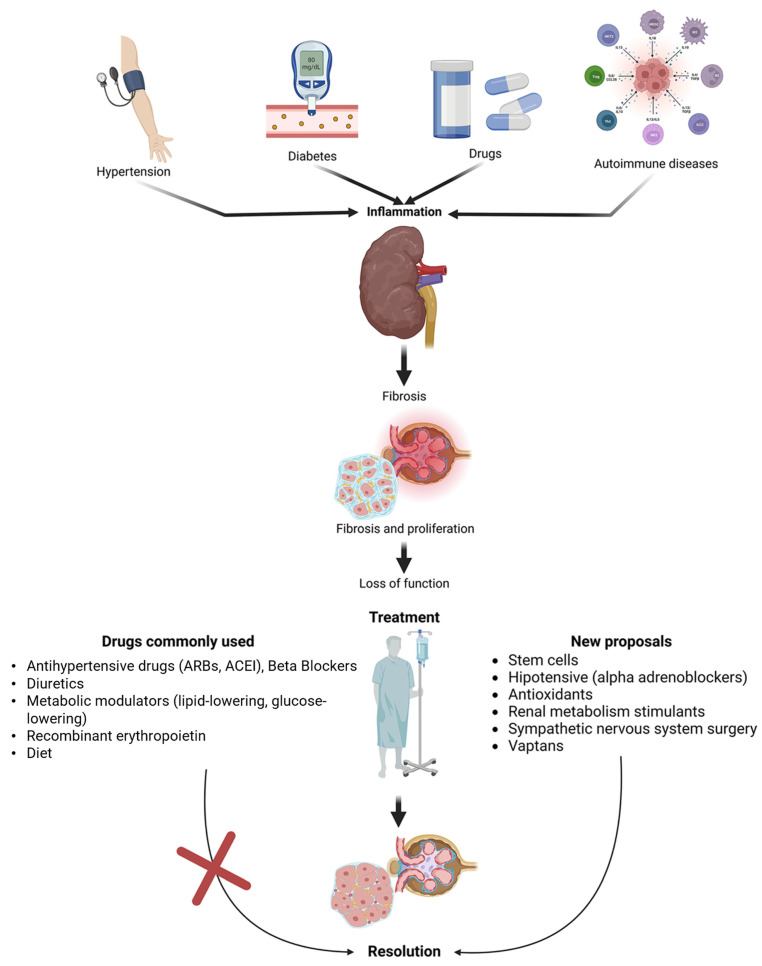
Graphical abstract: chronic kidney disease development and treatment; factors influencing renal injury, inflammation, extracellular matrix accumulation, fibrosis, and loss of function; commonly used drugs and novel proposals (figure created in BioRender.com accessed on 4 November 2022).

**Table 1 biomedicines-11-02828-t001:** Drugs commonly used in renal disease. GPD2: glycerol-3-phosphate (G3P) dehydrogenase; DPP-4: dipeptidyl peptidase 4; SGLT-2: Sodium-glucose cotransporter-2.

Therapeutic Target	Drugs	Mechanism	References
Antihypertensives	Alacepril Benazepril Captopril Cilazapril Delapril Enalapril Fosinopril Lisinopril Moexipril Perindopril Quinapril Ramipril Trandolapril Zofenopril	Angiotensin-converting enzyme inhibitors	[67,68,69,70,71,72,73,74]
	Irbesartan Losartan Telmisartán Valsartán	Angiotensin II receptor blockers	[61,62,75,76]
	Amlodipine Azelnidipine Cilnidipine Lercanidipine Felodipine	Calcium channel blockers	[77,78,79]
	Aliskirén	Renin-inhibitors	[80]
	Bisoprolol Carvedilol	β-blockerthird-generation hyb rid β-blocker	[70,77,81,82]
	Doxazosin	α-adrenolytic	
Diuretics	Epoxymexrenone 2nd Gen. Spironolactone 1st Gen. Canrenone 1st Gen. Eplerenone 2nd Gen.	Aldoserone receptor antagonist	[83,84]
	Furosemide	Na-K-Cl cotransporter antagonist	[85,86,87]
Hypoglycemic agents	Biguanides Metformin	Inhibit GPD2 to reduce hepatic gluconeogenesis.	[88,89]
	Sulfonylureas: Glibenclamide Gliclazid Glimepiride	Binding to ATP-dependent potassium channels in pancreatic β-cells.	[90,91]
	Vildagliptin Linagliptin	DPP-4 inhibitor	[92,93]
	Canaglifozin Empagliflozin	SGLT-2 inhibitor	[94,95,96,97]
Diet	Low-protein diet + Ketoanalogues Ketoskeril Protein-free rice		[98,99,100,101,102]
Vitamins and minerals	Potassium		[103,104,105]
	Iron		[106,107,108,109,110,111,112,113]
	Cholecalciferol		[114,115]
Calcimimetics	Cinacalcet	Parathyroid calcium-sensing receptor (CaR) binding compounds	[116,117]

**Table 2 biomedicines-11-02828-t002:** New treatment proposals for chronic kidney diseases.

Terapeutic Targets	Treatments	Mechanism	Clinical Trials	References
Cell therapy	Adipose mesenchymal tissue stem cells	-A Randomized, Controlled, Dose-Escalation. Pilot Study to Assess the Safety and Efficacy of a Single Intravenous Infusion of Allogeneic Mesenchymal Precursor Cells (MPCs) in Subjects with Diabetic Nephropathy and Type 2 Diabetes -Evaluation the Effect of Mesenchymal MSCs Transplantation in Patients with Chronic Renal Failure Because of Autosomal Dominant Polycystic Kidney Disease	NCT01843387 (2015) NCT02166489 (2016)	[162,163]
Metabolic modulators	Roziglitazone Pioglitazone	PPR-γ receptor agonist	NCT00169923 (2001–2005) NCT00174993. (2001–today)	[164,165,166,167]
Vasopressin inhibitors	-Tolvaptan	Vasopressin V_2_ receptor antagonist	NCT02331680 (2014–2016)	[168,169,170,171,172]
Antihypertensives	-Sitaxentan	A selective antagonist of endothelin 1 receptors ET-A/ET-B	NCT00810732 (2007–2009) NCT00160225	[173,174,175,176]
	-Recombinant Ang 1–7	Angiotensin II inhibitor	(Animal model)	[177]
Antifibrotics	-Selonsertib	Selective ASK1 inhibitor (effect of selonsertib (formerly GS-4997) on estimated glomerular filtration rate (eGFR) decline in participants with CKD).	NCT02177786 (2014–2018)	[178]
	-Enfuvirtide	G-protein-coupled chemokine receptor (CXCR4) antagonist.	(Animal model)	[179]
Antioxidants	-Icariin -Quercetina -Broccoli (Sulforaphane)-Resveratrol -Hydrogen Sulfide -Bardexolone -(BSHX)	Flavonoid Flavonoid Antioxidant Inhibition of NF-κB Nrf2 activation Ang II-mediated regulation of Na+ -K+ ATPase. Efficacy of Bardoxolone in chronic kidney disease (CKD) patients with type 2 diabetes	(Animal model) (Animal model) (Animal model) NCT03597568 (2019-today) (In vitro and in vivo animal models) NCT02316821 (2014–2017)	[180,181,182,183,184,185]
	-Propolis	Suppress α-SMA expression inhibits SMAD 2/3-dependent and independent pathways	(Animal model)	[186]
Retinoic acid	Vitamin A	Activation of signaling pathways (reprodution and organogenesis)	(Animal model)	[187]
Oral carbone absorbent (AST-120)	-Renamezin -Kremezin	Low molecular weight compounds absorption	NCT00500682 (2007–2011) and NCT00501046 (2007–2011)	[188,189,190]
Hypoglycemic agents	-Glucagon-like peptide-1 receptor agonist (GLP-1RA)	The Effect of Glucagon-like-peptide 1 (GLP-1) Receptor Agonism on Diabetic Kidney Disease	NCT01847313 (2013–2015)	[191,192,193,194]
Sympathetic nervous system antagonists	Surgery	Catheter denervation	NCT01832233 (2013–2015)	[195,196]
Demethylating agents	5-azacytidine Decitabine	DNA methylation inhibitor	(Animal trial, prospective)	[197,198,199]

## Data Availability

No new data were created.

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
