# Peer review of "Novel Approaches in Chronic Renal Failure without Renal Replacement Therapy: A Review"

_biomedicines, 2023, doi:10.3390/biomedicines11102828_

Round 1

Reviewer 1 Report

First sentence of introduction is nonsensical.

line 37-38   I do not see the need for mentioning paper where the study was published (although the name of the journal carries additional weight). I think mentioning the year is enough to see how up to date they are.

line 34    I do not see the need for capital letters in Acute Kidney Disease.

line 71  I do not understand the following sentence. "Recently has been informed the presence..."

section 2  I think diabetes mellitus deserves the priority.

line 114    There are two and in the sentence and I am not sure if the both are necessary. 

line 127    There are many types of hypertension and diabetes so I think to avoid the confusion it should be made clear what conditions are we referring to. 

Table 1.   

I do not see lercanidipine mentioned. 

Carvedilol Doxazosin are mentioned in the same line. 

Vitamins y minerals (I do not understand in the phrase).

I am not sure how erythropoietin recombinant fits in this category (vitamins and minerals).

line 137   Again what hypertension. You can mention arterial hypertension with abbreviation AH and use the abbreviation in the rest of the manuscript.

line 142  Same as hypertension. Instead of diabetes use abbreviation DM for diabetes mellitus.

line 143-144   "These drugs are prescribed to 22–33% of the general population and are useful..." Seems more natural formulation. 

line 154   Why capital letter for ramipril?

line 165   I do not see the explanation for AHT but I suppose it is arterial hypertension. If it is it needs to be mentioned earlier and I suggest AH instead of AHT.

line 177  RAAS abbreviation was used earlier so I suggest the continuation of use. 

line 194   Aliskiren with capital letter.

Verapamil and diltiazem are not mentioned in the previous table but are suggested as good for use. Needs further explanation. 

Bisoprolol is not even mentioned in the adenoblockers section.

line 216   T2DM abbreviation was used before. Use only abbreviation after its initial explanation.

line 250 Glibenclamide, gliclazide, and glimepiride are all antidiabetic drugs from the same group (sulfonylureas). 

line 274  "is" seems unecessary.

line 278   Again T2DM instead of type 2 diabetes.

line 284   Reformulate "evidenced".

line 286  "is" is misplaced.

              "... as specific"

line 290   Renoprotective is one word.

line 291-292   "of the utmost importance..."

line 292   "patients with CKD"

line 296  I suppose "SGLT2 is" is misspelling of SGLTis.  

Diet section needs simplification. It is difficult to separate what are the recommendation differences in pre-dialysis and dialysis group. 

Also, reduction in food rich in potassium or at least differences in preparation of food rich in potassium should be more emphasized.  

line 139-140    ACE inhibitors as ACEIs and angiotensin II receptor antagonists as ARBs.

Introduction to the main point of the article seems too long.

I am not sure if cinacalcet is novel therapy. It is in use for twenty years. Same goes for MRAs.

line 663-672   I do not understand the need for this paragraph.

Renal sympathetic denervation is also in use for quite some time now.

Future direction section could be simplified/shortened.

Figure 1.   

Hypotensive?

Recombinant erythropoietin instead of just erythropoietin.

line 747    slowing or stopping 

line 748-749   Renal replacement therapy may be better term than "dialysis or renal transplantation".

line 751   ARA-II is same as ARBs if I am not mistaken.

line 802 "better control" instead of "more efficiently controlling"

There are too many references.

Article needs extensive language editing.

Author Response

Response to Reviewers

 First of all, we would like to thank the reviewers and editor for their recommendations. Below, we provide a point by point response to all comments raised by the reviewers.

Reviewer 1

  1. First sentence of introduction is nonsensical.

Response: Your suggestion has been taken into account and the first sentence of the introduction has been eliminated (line 19)

  1. Line 37-38 I do not see the need for mentioning paper where the study was published (although the name of the journal carries additional weight). I think mentioning the year is enough to see how up to date they are.

Response: We agree with the reviewer, we removed where the study was published, and left only the year.

  1. line 34 I do not see the need for capital letters in Acute Kidney Disease

Response: The capital letters of Acute Kidney Disease have been removed.

  1. line 71 I do not understand the following sentence. "Recently has been informed the presence..."

Response: The sentence on line 71 has been restructured for better understanding.

  1. section 2 I think diabetes mellitus deserves the priority.

Response: We agree with your comment, this section has been enriched with new topics that you will find in sections 2.3, 2.4 and 2.5.

  1. line 114 There are two and in the sentence and I am not sure if the both are necessary

Response: Thank you for your comment, we have reviewed the grammar and it has been modified.

  1. line 127 There are many types of hypertension and diabetes so I think to avoid the confusion it should be made clear what conditions are we referring to

Response: We refer to arterial hypertension, the change made can be seen in line 157.

  1. Table 1. I do not see lercanidipine mentioned.

Response: Thanks for the suggestion, the drug has been included in table 1.

  1. Carvedilol Doxazosin are mentioned in the same line.

Response: Correction has been made

  1. Vitamins y minerals (I do not understand y in the phrase)

Response: Correction has been made

  1. I am not sure how erythropoietin recombinant fits in this category (vitamins and minerals).

Response: We agree with you, and indeed erythropoietin does not correspond to this topic.

  1. Line 137 Again what hypertension. You can mention arterial hypertension with abbreviation AH and use the abbreviation in the rest of the manuscript

Response: The suggestion has been taken into account and corrected throughout the text.

  1. line 142 Same as hypertension. Instead of diabetes use abbreviation DM for diabetes mellitus

Response: Thank you for your comment, the correction has been made.

  1. line 143-144 "These drugs are prescribed to 22–33% of the general population and are useful..." Seems more natural formulation

The wording of the sentence has been improved for better understanding.

  1. line 154 Why capital letter for ramipril?

Response: Correction has been made.

  1. line 165 I do not see the explanation for AHT but I suppose it is arterial hypertension. If it is it needs to be mentioned earlier and I suggest AH instead of AHT

Response: We agree with you, AHT refers to arterial hypertension, and following your suggestion we have changed AHT to AH and included the explanation of the abbreviation.

  1. Line 177 RAAS abbreviation was used earlier so I suggest the continuation of use

Response: Thank you for the suggestion

  1. line 194 Aliskiren with capital letter.

Response: thank you for your comment, the correction has been made (line 234).

  1. Verapamil and diltiazem are not mentioned in the previous table but are suggested as good for use. Needs further explanation

Response: The drugs have been included in the table; in addition, a better explanation has been provided.

  1. Bisoprolol is not even mentioned in the adenoblockers section.

Response: Thanks for the comment, bisoprolol has been included in the adrenoblockers section, and in Table 1.

  1. line 216 T2DM abbreviation was used before. Use only abbreviation after its initial explanation.
  2.  

Response: Correction has been made in the text.

  1. line 250 Glibenclamide, gliclazide, and glimepiride are all antidiabetic drugs from the same group (sulfonylureas).

Response: Thanks for the comment, it has been corrected and is on line 281.

  1. line 274 "is" seems unecessary.

Response: The “is” has been removed (line 301)

  1. line 278 Again T2DM instead of type 2 diabetes.

Response: Thanks for the comment, it has been corrected throughout the text.

  1. line 284 Reformulate "evidenced".

Response: The sentence has been restructured and the word “evidenced” has been removed for better understanding.

  1. line 286 "is" is misplaced and line 286, and as a specific"

Response: The error has been corrected (line 306)

  1. line 290 Renoprotective is one word.

Response: Thanks for the comment, the correction has been made in the text.

  1. line 291-292 "of the utmost importance..."

Response: The “utmost importance” has been included (line 315).

  1. line 292 "patients with CKD"

Response: Thanks for the comment, it has been included in line 315.

  1. line 296 I suppose "SGLT2 is" is misspelling of SGLTis.

Response: Error was corrected (line 319)

  1. Diet section needs simplification. It is difficult to separate what are th recommendation differences in pre-dialysis and dialysis group. Also, reduction in food rich in potassium or at least differences in preparation of food rich in potassium should be more emphasized.

Response: The paragraph has been restructured for better understanding and the importance of sodium and potassium in the diet has been emphasized.

  1. line 139-140 ACE inhibitors as ACEIs and angiotensin II receptor antagonists as ARBs, Introduction to the main point of the article seems too long.

Response: The paragraph has been restructured.

  1. I am not sure if cinacalcet is novel therapy. It is in use for twenty years. Same goes for MRAs.

Response: Cinacalcet, and calcimimetics have been relocated to Table 1 and in the manuscript in section 3.2.5 respectively. MRAs were also relocated in the manuscript (section 3.4).

  1. line 663-672 I do not understand the need for this paragraph.

Response: The paragraph has been removed as it is not related to the CKD, as mentioned by the reviewer.

  1. Figure 1. Hypotensive?, and Recombinant erythropoietin instead of just erythropoietin.

Response: Thanks for the comment, corrections have been made to the figure.

  1. line 747 slowing or stopping

Response: word has been changed to stopping

  1. line 748-749 Renal replacement therapy may be better term than

"dialysis or renal transplantation"

Response: Gracias por la observación, se incluyó el termino Renal replacement therapy (line 736).

  1. line 751 ARA-II is same as ARBs if I am not mistaken.

Response: The term ARA-II was not included in section 5 (future direction), it was necessary to summarize the information in this section.

  1. line 802 "better control" instead of "more efficiently controlling"

Response: The word "better control" was included in the conclusions section.

  1. There are too many references.

Response: We thank the reviewer for the comment, we consider that the bibliography used is necessary to support the information in the manuscript, we also commented to the reviewer that more references have been included at the request of reviewer 2.

  1. Article needs extensive language editing.

Response: On your recommendation, we have sent the manuscript for linguistic review with an English language proofreader and editor (certified medical writer), thank you for your comment.

Reviewer 2 Report

Review of the manuscript biomedicines-2509730

Novel Approaches in Chronic Renal Failure without Renal Re- placement Therapy: A Review

By Martínez-Hernández SL et al.

GENERAL NOTE: Unfortunately, the manuscript adds nothing new to the current treatment of chronic kidney disease (CKD). Also, the description of the novel and future prospects of treatment is very fragmentary. I understand that the Authors have attempted to synthesize the content on the CKD pathophysiology and systematize the issue of CKD treatment, which could have educational value for many readers. However, the manuscript in its current form is in many places too superficial. Definitely needs redrafting and reorganizing the order of the content presented.

MAJOR REMARKS

1. Throughout the text, and especially in chapter 4, the authors refer enigmatically to many clinical trials without giving their names (acronyms) and the range of years during which the clinical trials were conducted.

2. At the end of the "Introduction" there should be a fragment on the purpose of the review, along with the premises justifying the discussion of this topic. In addition, the methodology of the non-systematic review performed should be briefly described.

3. Placing the subchapter "Pathophysiology of fibrosis" which describes this process in detail, within the Introduction, is incomprehensible. I believe that this section should appear in the manuscript after discussing hypertension and diabetes as the main causes of CKD, because fibrosis is a consequence of complex pathophysiological disorders occurring in the kidneys in the course of hypertensive / diabetic nephropathy.

4. Section 2.1. The description of the RAA system is very simplified. Other components of the RAA system, type II converting enzyme, angiotensin 1-7, angiotensin III and IV, are now known. In addition, the RAA system is currently perceived as not an isolated system, but a component of a larger protease system - the RAA is "associated" with the system of kinins, natriuretic peptides, endothelin, etc. This description must be definitely reworded and extended.

5. Section 2.2. Diabetes mellitus - microangiopathic complications – the described pathomechanism  is very poor. There are other pathophysiological issues besides oxidative stress involving in pathogenesis of microangiopathic disorder, e.g. activation of the polyol pathway, formation of late glycation end products, activation of the hexosamine pathway, activation of protein kinase C, release of numerous inflammatory mediators etc.

6. The drugs listed in Table 1 are incomplete; these are only selected examples and not a complete catalogue. The table lists unexplained abbreviations (DPP-4; SGLT-2). The indicated mechanism of action of metformin is fragmentary. Carvedilol is a third-generation hybrid beta-blocker; doxazosin is an alpha-blocker. Spironolactone - there are several other, newer aldosterone antagonists.

7. Chapter 3 – in the current procedure of CKD treatment, it is also worth mentioning the reduction of atherosclerosis risk factors, such as obesity, hypercholesterolemia and smoking. It is also worth mentioning the avoidance of drugs with a high potential for nephrotoxicity, e.g.

https://www.ncbi.nlm.nih.gov/pmc/articles/PMC9960203/

https://www.ncbi.nlm.nih.gov/pmc/articles/PMC3794522/

8. Chapter - Novel proposals - the classes of drugs listed in Table 2 are then discussed too vaguely later in the text. In many places, e.g. 4.3.2. Treatments based on Angiotensin 1-7 inhibitor; 4.6. Retinoic acid (RA) (vitamin A); 4.8. Glucagon-like peptide-1 receptor agonist (GLP-1RA); there is no mention of any research justifying the use of a given group of drugs in CKD.

9. Table 2. Should be redrafted - with the names of clinical trials / short description of preclinical studies supporting the use in CKD for a given class, along with the relevant references.

10. Antioxidant treatment is currently the most advanced direction of CKD treatment and this issue should be described much more broadly, taking into account, e.g.:

 https://www.termedia.pl/Oxidative-stress-mechanisms-as-potential-therapeutic-targets-in-chronic-kidney-disease,67,47380,1,1.html

http://www.ijkd.org/index.php/ijkd/article/view/2044/766

11. How do the compounds described in Section 7 New Approaches to Electrolyte Management and Section 8 Future Directions differ from those in Section 4, since the compounds in Section 4 are also not currently used in common clinical practice?

12. Why are the compounds described in chapter 6 6. Demethylating agents separated in the form of a subchapter, since they are also part of the "novel", future treatment?

13. The study also does not address the issue of modulating intestinal dysbiosis (e.g. using probiotics, prebiotics or synbiotics) as a promising and increasingly used direction in the treatment of CKD.

MINOR REMARKS

1. Sentence line 31 – incomprehensible

2. Introduction – Acute Kidney Disease – or – Acute Kidney Injury?

3. Lines 40-41 - Definition of CKD should be clarified - what decrease in GFR warrants the diagnosis of CKD?

4. CKD advancement based on GFR value - you should use "G" in place of "level"; G1-G5

5. The list of References has not been prepared in accordance with the guidelines of the Journal - in terms of providing authors, bibliographic data of cited journals, etc.

Author Response

Response to Reviewers

Reviwer 2

We appreciate the extensive review of the manuscript by the reviewer, which has allowed us to redesign, improve and deepen the content of the topics addressed, the specific points suggested and to include the topics requested by the reviewer.

MAJOR REMARKS

  1. Throughout the text, and especially in chapter 4, the authors refer enigmatically to many clinical trials without giving their names (acronyms) and the range of years during which the clinical trials were conducted.

Response: We have described the details of the names and dates of all clinical trials.

  1. At the end of the "Introduction" there should be a fragment on the purpose of the review, along with the premises justifying them discussion of this topic. In addition, the methodology of the non-systematic review performed should be briefly described.

Response: We have included at the end of the introduction the purpose and methodology developed for this non-systematic review.

  1. Placing the subchapter "Pathophysiology of fibrosis" which describes this process in detail, within the Introduction, is incomprehensible. I believe that this section should appear in the manuscript after discussing hypertension and diabetes as the main causes of CKD, because fibrosis is a consequence of complex pathophysiological disorders occurring in the kidneys in the course of hypertensive / diabetic nephropathy.

Response: Thanks for the suggestion, the Pathophysiology of fibrosis has been enriched and relocated after hypertension (section 2.6).

  1. Section 2.1. The description of the RAA system is very simplified. Other components of the RAA system, type II converting enzyme, angiotensin 1-7, angiotensin III and IV, are now known. In addition, the RAA system is currently perceived as not an isolated system, but a component of a larger protease system - the RAA is "associated" with the system of kinins, natriuretic peptides, endothelin, etc. This description must be definitely reworded and extended.

Response: Following the reviewer's suggestions we have reworded and extended the description the RAA system.

  1. Section 2.2. Diabetes mellitus - microangiopathic complications the described pathomechanism is very poor. There are other pathophysiological issues besides oxidative stress involving in pathogenesis of microangiopathic disorder, e.g. activation of the polyol pathway, formation of late glycation end products, activation of the hexosamine pathway, activation of protein kinase C, release of numerous inflammatory mediators etc.

Response: The section has been enriched taking into account the reviewer's comments, including: pathogenesis of the microangiopathic disorder, activation of the thepolyol pathway, formation of late glycation end products, activation of the hexosamine pathway, activation of protein kinase C, release of numerous inflammatory mediators.

  1. The drugs listed in Table 1 are incomplete; these are only selected examples and not a complete catalogue. The table lists unexplained abbreviations (DPP-4; SGLT-2). The indicated mechanism of action of metformin is fragmentary. Carvedilol is a third generation hybrid beta-blocker; doxazosin is an alpha-blocker. Spironolactone – there are several other, newer aldosterone antagonists.

Response: Table 1 has been completed and updated with new drugs, and the abbreviations used are also explained.

  1. Chapter 3 – in the current procedure of CKD treatment, it is also worth mentioning the reduction of atherosclerosis risk factors, such as obesity, hypercholesterolemia and smoking. It is also worth mentioning the avoidance of drugs with a high potential for nephrotoxicity, e.g.

Response: Chapter 3.1 has been updated to include the points noted by the reviewer.

  1. Chapter - Novel proposals - the classes of drugs listed in Table 2 are then discussed too vaguely later in the text. In many places, e.g. 4.3.2. Treatments based on Angiotensin 1-7 inhibitor; 4.6. Retinoic acid (RA) (vitamin A); 4.8. Glucagon-like peptide-1 receptor agonist (GLP-1RA); there is no mention of any research justifying the use of a given group of drugs in CKD.

Response: Table 2 has been restructured, drugs that are not related to CKD have been removed, and the points noted (section 4) have been addressed.

  1. Table 2. Should be redrafted - with the names of clinical trials / short description of preclinical studies supporting the use in CKD for a given class, along with the relevant references.

Response: Table 2 has been restructured and the names of the clinical trials have been included, as well as a brief description of preclinical studies supporting the use in CKD.

  1. Antioxidant treatment is currently the most advanced direction of CKD treatment and this issue should be described much more broadly.

Response: The section on antioxidants was updated and new information was included taking into account the articles proposed by the reviewer.

  1. How do the compounds described in Section 7 New Approaches to Electrolyte Management and Section 8 Future Directions differ from those in Section 4, since the compounds in Section 4 are also not currently used in common clinical practice?

Response: Unified the chapters, restructured the paragraph for better understanding (section 3.6).

  1. Why are the compounds described in chapter 6 6. Demethylating agents separated in the form of a subchapter, since they are also part of the "novel", future treatment?

Response: The demethylating agents were relocated at the end of chapter 4.9.

  1. The study also does not address the issue of modulating intestinal dysbiosis (e.g. using probiotics, prebiotics or synbiotics) as a promising and increasingly used direction in the treatment of CKD.

Response: Thanks for the comment, a new section has been included in the manuscript that will address probiotics (section 4.1).

MINOR REMARKS

  1. Sentence line 31 – incomprehensible

response: Corrected meaningless phrases in the manuscript and revised the information in the section for better understanding, line 31 was corrected.

  1. Introduction – Acute Kidney Disease – or – Acute Kidney Injury?

Response: We decided to use the term Acute Kidney Disease.

  1. Lines 40-41 - Definition of CKD should be clarified

Response: gracias por su comentario hemos modificado la definición de CKD para su mejor entendimiento.

  1. what decrease in GFR warrants the diagnosis of CKD?

Response:  According to clinical practice guideline update for the diagnosis, evaluation, prevention, and treatment of chronic kidney disease, (KDIGO (2017), CKD is considered when the eGFR is below < 60 ml/min/1.73 m2

  1. CKD advancement based on GFR value - you should use "G" in place of "level"; G1-G5

Response: It was included in line 27-30

  1. The list of References has not been prepared in accordance with the guidelines of the Journal - in terms of providing authors, bibliographic data of cited journals, etc.

Response: Thanks for the comment, the citations have been reordered following the magazine's guidelines.

Round 2

Reviewer 1 Report

It is not clear how from 36,000 papers 240 were included. This requires further explanation.

Also, I do not understand why search word was not "Chronic kidney disease" instead of "Chronic", "Kidney" and "Disease". In MeSH the search word is "Renal Insufficiency, Chronic". 

It is PubMed instead of PUBMED.

line 76 - Arterial hypertension instead of Hypertension

line 82 - vasodilatation instead of vasodilation

line 84 - Again "vasodilation" is used.

line 110 - Diabetes mellitus (DM)...

line 153 - DM instead of "diabetes"

Section Principles etiologies of CKD seems out of place, too long and almost unecessary.  

line 205-206 - CKD often goes unrecognized without any symptoms.

General considerations paragraph, especially the first one, is written inconsistent and too complicated.

The whole article requires extensive revision.

Table 1. - Aldosterone Receptor Antagonist; Glibenclamide etc. are Sulfonylureas... too long explanations for metformin and sulfonylureas

line 249 - "a" is not necessary.

There is no meaningful order in presented data. It is evident that a lot of research was done but the data presented are incoherent. 

line 321 - Why capital letter in "Lercanidipine"?

Figure 1. - Hypotensors? Antihypertensive drugs is more appropriate.

Article requires extensive language editing.

Author Response

REVIEWER 1

Comments and Suggestions for Authors

It is not clear how from 36,000 papers 240 were included. This requires further explanation.

Also, I do not understand why search word was not "chronic kidney disease" instead of "Chronic", "Kidney" and "Disease". In MeSH the search word is "Renal Insufficiency, Chronic".

Response: to the first point, we used the PMC advanced search builder query, not as a simple google search:

https://www.ncbi.nlm.nih.gov/pmc/?term=((((((((((Chronic)+AND+Renal)+AND+Failure)+AND+Treatment)+AND+adult))+AND+(%222010%2F12%2F01%22%5BEntrez+Date%5D+%3A+%222023%22%5BEntrez+Date%5D)))+NOT+Meta-anaysis)+NOT+Trial)+NOT+Systematic

In addition, following suggestions from other reviewers in the first round, it was pointed out to us that for a better explanation it was necessary to include some clinical trials highlighting important points of the treatment throughout the paper.

To the second point, we do not use MeSH database, only PMC.

It is PubMed instead of PUBMED.

Response: according with you, we corrected the word to PubMed.

line 76 - Arterial hypertension instead of Hypertension

Response: accordingly

line 82 - vasodilatation instead of vasodilation

Response: done

line 84 - Again "vasodilation" is used.

Response: done

line 110 - Diabetes mellitus (DM)...

Response: we agree

line 153 - DM instead of "diabetes"

Response: done

Section Principles etiologies of CKD seems out of place, too long and almost unnecessary.  

line 205-206 - CKD often goes unrecognized without any symptoms.

Response: we have been taken in consideration your valuable suggests

General considerations paragraph, especially the first one, is written inconsistent and too complicated.

Response: the paragraph has been reconstructed for better understanding

The whole article requires extensive revision.

Table 1. - Aldosterone Receptor Antagonist; Glibenclamide etc. are Sulfonylureas... too long explanations for metformin and sulfonylureas

Response: we agree, so we rewrite table 1

line 249 - "a" is not necessary.

Response: we agree

There is no meaningful order in presented data. It is evident that a lot of research was done but the data presented are incoherent. 

Response: We thank you for your comment, we are aware that this version of the paper has changed with respect to the version sent previously, this is because the reviewers of the first round made some punctual and very valuable observations that have already been solved. It should be noted that reviewer 2 of the first round suggested a reordering and addition of new topics for a better understanding of the manuscript, so this second version after a complete revision, we consider that it already provides a clearer picture of the updated treatment of this complex disease.

line 321 - Why capital letter in "Lercanidipine"?

Response: we agree

Figure 1. - Hypotensors? Antihypertensive drugs is more appropriate.

Response: We correct figure 1

Comments on the Quality of English Language

Article requires extensive language editing.

Response: Professional English proofreading was done; certificate of proofreading is submitted.

Reviewer 2 Report

Re-review of the manuscript biomedicines-2509730

Novel Approaches in Chronic Renal Failure without Renal Re-2 placement Therapy: A Review

Sandra Luz Martínez-Hernández, Martín Humberto Muñoz-Ortega, Manuel Enrique Ávila-Blanco, Mariana Yazmin Medina-Pizaño  and Javier Ventura-Juárez

Thank you for submitting another updated version of the manuscript. Also, thank you for considering my suggestions. I believe that in its current version, the manuscript has gained much in terms of comprehensiveness and clarity. Before the final publication, the authors could still remove minor defects, including:

- Table 1 and text of the manuscript - instead of the term "calcium channels inhibitors" it is better to use the term "calcium channels blockers". In addition, verapamil and diltiazem, as non-dihydropyridine calcium channel blockers, are generally not used as hypotensive drugs, but they are used in the pharmacotherapy of ischemic heart disease and as antiarrhythmic drugs

- Table 1 - doxazosin - the drug should be listed on a separate line as alpha-adrenolytic

- Table 1 - is felodipine also a calcium channel blocker or a duretic-aldosterone antagonist?

- Table 2 - no column headers??? Should be added

- Table 2 - is losartan really an ET-A / ET-B receptor antagonist? Or should other "sentans" like bosentan and others be mentioned in this verse?

Table 2 - "recombinant Ang 1-7 - is it really an "Ag II inhibitor"? Maybe it's better to just attribute to this compound is a vasodilator agents that antagonizes the pressor effect of AgII?

Author Response

REVIEWER 2

Comments and Suggestions for Authors

Re-review of the manuscript biomedicines-2509730

Novel Approaches in Chronic Renal Failure without Renal Re-2 placement Therapy: A Review

Sandra Luz Martínez-Hernández, Martín Humberto Muñoz-Ortega, Manuel Enrique Ávila-Blanco, Mariana Yazmin Medina-Pizaño  and Javier Ventura-Juárez

Thank you for submitting another updated version of the manuscript. Also, thank you for considering my suggestions. I believe that in its current version, the manuscript has gained much in terms of comprehensiveness and clarity. Before the final publication, the authors could still remove minor defects, including:

- Table 1 and text of the manuscript - instead of the term "calcium channels inhibitors" it is better to use the term "calcium channels blockers". In addition, verapamil and diltiazem, as non-dihydropyridine calcium channel blockers, are generally not used as hypotensive drugs, but they are used in the pharmacotherapy of ischemic heart disease and as antiarrhythmic drugs.

Response: The correction was accepted; verapamil and diltiazem were dropped from the table.

- Table 1 - doxazosin - the drug should be listed on a separate line as alpha-adrenolytic

- Table 1 - is felodipine also a calcium channel blocker or a diuretic-aldosterone antagonist?

Answer: It is a calcium blocker; the requested modification has been made.

- Table 2 - no column headers??? Should be added.

Answer: Headers have been added.

- Table 2 - is losartan really an ET-A / ET-B receptor antagonist? Or should other "sentans" like bosentan and others be mentioned in this verse?

Answer: The requested change was made; it corresponds to table 1 as angiotensin II inhibitor.

Table 2 - "recombinant Ang 1-7 - is it really an "Ag II inhibitor"? Maybe it's better to just attribute to this compound is a vasodilator agents that antagonizes the pressor effect of AgII?

Round 3

Reviewer 1 Report

line 343-346: Should be two sentences. 

Once again, it seems a lot of research was done but the article still seems incoherent. 

There needs to be some order in paragraphs. Pathophysiology, existing evidence, benefit of the treatment in population with CKD etc. 

There is a lot of information in the article but it is hard to draw valuable conclusions since there is no order in writing. One sentence follows another with no clear connection. 

Moderate English editing is required.

Author Response

Letter to response reviewer 1 Third round                    09/27/2023

First of all, we would like to thank the reviewer and editor for their recommendations. Below, we provide a point by point response to all comments raised by the reviewer 1.

line 343-346: Should be two sentences

Response: The correction has been made.

Once again, it seems a lot of research was done but the article still seems incoherent. 

There needs to be some order in paragraphs. Pathophysiology, existing evidence, benefit of the treatment in population with CKD etc. 

Response: a new paragraph sequence analysis has been carried out and we have tried to organize the sections in a more coherent way in the manuscript, we also add the suggested changes

There is a lot of information in the article but it is hard to draw valuable conclusions since there is no order in writing. One sentence follows another with no clear connection. 

Response. Thanks for the suggestion, we have tried to make the paragraphs in the manuscript have a clear and understandable connection.

Comments on the Quality of English Language

Moderate English editing is required.

Response: the manuscript was reviewed in the English Language

Javier Ventura-Juárez

Correspondance author.